# SPARTA ALIGNMENT: Collectively Aligning Multiple Language Models through Combat

**Yuru Jiang**[*1]    **Wenxuan Ding**[*2]    **Shangbin Feng**[*3]    **Greg Durrett**[2]    **Yulia Tsvetkov**[3]

[1]Zhejiang University    [2]New York University    [3]University of Washington

3220102689@zju.edu.cn   wd2403@nyu.edu   shangbin@cs.washington.edu

## Abstract

We propose SPARTA ALIGNMENT, an algorithm to collectively align multiple LLMs through competition and combat. To complement a single model's lack of diversity in generation and biases in evaluation, multiple LLMs form a "sparta tribe" to compete against each other in fulfilling instructions while serving as judges for the competition of others. For each iteration, one instruction and two models are selected for a duel, the other models evaluate the two responses, and their evaluation scores are aggregated through a adapted elo-ranking based reputation system, where winners/losers of combat gain/lose weight in evaluating others. The peer-evaluated combat results then become preference pairs where the winning response is preferred over the losing one, and all models learn from these preferences at the end of each iteration. SPARTA ALIGNMENT enables the self-evolution of multiple LLMs in an iterative and collective competition process. Extensive experiments demonstrate that SPARTA ALIGNMENT outperforms initial models and 4 self-alignment baselines across 10 out of 12 tasks and datasets with 7.0% average improvement. Further analysis reveals that SPARTA ALIGNMENT generalizes more effectively to unseen tasks and leverages the expertise diversity of participating models to produce more logical, direct and informative outputs.[†]

## 1 Introduction

Aligning large language models (LLMs) has become a critical step of LLM post-training, steering LLMs for specialized skills [25, 28], helpful responses [2], and alignment with human values [37, 10, 85]. Existing alignment recipes would require external signals to learn from (e.g., reward models [40, 60, 7] or preference pairs [81, 40, 71, 64]) while high-quality supervision is often scarce and costly [2, 68]. As such, recent research investigates *self-alignment*, where the LLM itself serves as its own judge [50], reward model [96, 14], or supervision signal [49, 81], without any external information. These approaches have demonstrated gains on instruction following and reasoning problems, initiating a paradigm of self-improving language models [105, 104, 61, 95, 9].

However, we posit that the recipe of "one model aligning itself" might not be reliable for two reasons:

- *A single model struggles to reliably judge its own generation*. There is ample evidence on LLM self-bias [101, 31, 93], where one model consistently favors its own response and reinforces its own priors/biases in LLM-as-a-judge evaluations [103]. Impact of the biased self-judgment will be even more pronounced for tasks and instructions with underpinning human values and cultural implications [78, 67, 73], propagating and reinforcing the flaws.

- *A single model struggles to reliably generate diverse responses to learn from*. RL and preference learning recipes benefit from a wide spectrum of responses with varying quality and reward

---

[*]equal contribution

[†]Resources available at https://github.com/yurujiang2003/sparta.

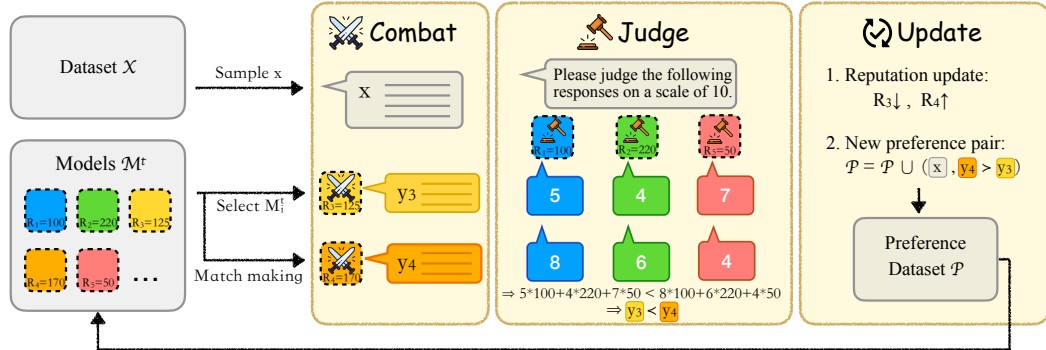

Figure 1: Overview of SPARTA ALIGNMENT, an algorithm collectively aligning multiple models via combat. SPARTA ALIGNMENT requires a dataset $\mathcal{X}$ and a pool of models $\mathcal{M}^t$. For each iteration $t$, we repeatedly sample a prompt $x$ from $\mathcal{X}$. For $x$, we first select a model $M_i^t$ and then select its opponent $M_{i'}^t$ with our Match-Making Strategy ($i = 3, i' = 4$ in the example). We then employ three steps: (1) Combat: the model pair generate responses $y_i, y_{i'}$ to $x$. (2) Judge: other models in the pool act as judges, generating their scores to $y_i, y_{i'}$. (3) Update: update the reputation scores of $M_i^t$ and $M_{i'}^t$ based on scores from Judge phase and create a new preference for the preference dataset $\mathcal{P}$. At the end of each iteration, we align models via preference learning on $\mathcal{P}$.

scores [102]. A single model, despite sampling, mostly generates homogeneous responses with similar styles and failure patterns [80, 105], struggling to provide the diversity and distinguishability of responses for robust preference learning.

Therefore, the single model itself becomes the bottleneck for self-evolution beyond its training priors, existing strengths and weaknesses, as well as its inherent biases.

To this end, we propose SPARTA ALIGNMENT, an algorithm to collectively align multiple language models. To complement each model's limitation in *judging* and *generating*, multiple LLMs form a "sparta tribe" to compete against each other [55, 98, 17] at fulfilling instructions and at the same time serve as judges for other LLMs. Specifically, SPARTA ALIGNMENT takes a pool of LMs $\mathcal{M}^0$ and a set of instructions $\mathcal{X}$ as input. For each iteration $t$, we repeatedly sample an instruction $x \in \mathcal{X}$ and design a *match-making system* to select two models $M_i^t$ and $M_{i'}^t$ for "combat", i.e., dueling to better respond to $x$. All the other models $\mathcal{M}^t \setminus \{M_i^t, M_{i'}^t\}$ then judge their responses and the aggregated judgment determines who won the combat, resulting in a preference pair such as $(x, M_i^t(x) \prec M_{i'}^t(x))$. Motivated by game theory [19], the judgment aggregation is weighted by the *reputation scores* of models, where the winning models increase and the losing models decrease after combat. At the end of each iteration, all of $\mathcal{M}^t$ go through preference learning [76, 35] with the collection of preference pairs in this iteration. After iterations of SPARTA ALIGNMENT, a few models in $\mathcal{M}^t$ would emerge as the strongest with the highest reputation scores for inference and deployment. SPARTA ALIGNMENT uniquely enables the self-evolution [112] of LLMs in a collective competition process, alleviating the limitations of a single model.

Extensive experiments on 12 datasets show that SPARTA ALIGNMENT consistently improves LLMs across 10 of the 12 tasks, outperforming the strongest baseline by 7% on average. SPARTA ALIGNMENT achieves the highest generalization accuracy in 2 out of 3 categories on challenging mathematical benchmarks, demonstrating robust generalization [8] beyond $\mathcal{X}$. SPARTA ALIGNMENT resembles the social stratification theory [62, 63, 75], where models in the Sparta tribe show different "classes" of reputation scores at the end of alignment. Further analysis reveals that the "sparta tribe" benefits from having more members and that SPARTA ALIGNMENT benefits from a diverse set of models with different expertise as $\mathcal{M}$ for improved generation diversity and more distinguishable preference pairs.

## 2 Methodology

**Overview**    SPARTA ALIGNMENT assumes access to a pool of LLMs, denoted as $\mathcal{M}^0 = \{M_k^0\}_{k=1}^m$ where 0 denotes the 0-th iteration, and a dataset $\mathcal{X}$. The models evolve iteratively for $\mathcal{T}$ iterations. In

---

**Algorithm 1:** Sparta Alignment

---

1   **Input:** Initial models $\mathcal{M}^0 = \{M_i^0\}_{i=1}^m$, with attributes $R_i$, where $R_i$ is model's reputation dynamically updated by combat results, Dataset $\mathcal{X}$; Max iteration $\mathcal{T}$.

2   **for** $t = 1$ **to** $\mathcal{T}$ **do**

3     Preference dataset $\mathcal{P} \leftarrow \emptyset$

4     **for** $x \in \mathcal{X}$ **do**

5       Select a model $M_i^t$ from $\mathcal{M}^t$

6       Select its opponent $M_{i'}^t = \text{Match-Making}(M_i^t)$

7       Combat: both models generate a response to $x$, $M_i^t(x)$ and $M_{i'}^t(x)$

8       **for** $M_k^t \in \mathcal{M}^t \setminus \{M_i^t, M_{i'}^t\}$ **do**

9         Evaluate both responses: $s_i^{(k)} \leftarrow \mathcal{J}(x, M_i^t(x), M_k^t)$, $s_{i'}^{(k)} \leftarrow \mathcal{J}(x, M_{i'}^t(x), M_k^t)$, where $\mathcal{J}(x, M_i^t(x), M_k^t)$ denotes $M_k^t$ evaluating response $M_i^t(x)$

10       **end**

11       $\bar{s}_i = \text{Aggregate}(\{s_i^{(k)}\}, \{R_k\})$, $\bar{s}_{i'} = \text{Aggregate}(\{s_{i'}^{(k)}\}, \{R_k\})$, $M_k^t \in \mathcal{M}^t \setminus \{M_i^t, M_{i'}^t\}$

12       **if** $\bar{s}_i > \bar{s}_{i'}$ **then** new preference pair $\mathcal{P} \leftarrow \mathcal{P} \cup \{x, M_i^t(x) \succ M_{i'}^t(x)\}$

13       **else** new preference pair $\mathcal{P} \leftarrow \mathcal{P} \cup \{x, M_{i'}^t(x) \succ M_i^t(x)\}$

14       $R_i, R_{i'} \leftarrow \text{Reputation-Update}(R_i, R_{i'}, \bar{s}_i, \bar{s}_{i'})$

15     **end**

16     **for** $M_i^t \in \mathcal{M}^t$ **do**

17       $M_i^{t+1} \leftarrow \text{DPO}(M_i^t, \mathcal{P})$

18     **end**

19   **end**

20   **Output:** Better Models $\mathcal{M}^{\mathcal{T}} = \{M_i^{\mathcal{T}}\}_{i=1}^m$

---

iteration $t$, we repeatedly sample $x \in \mathcal{X}$, select two models $M_i^t, M_{i'}^t$ for head-to-head *combat* via our Match-Making System, and generate responses $M_i^t(x), M_{i'}^t(x)$. The other models $\mathcal{M}^t \setminus \{M_i^t, M_{i'}^t\}$ then act as judges and evaluate $M_i^t(x)$ and $M_{i'}^t(x)$ to produce scores. These scores are aggregated using a weighted scheme based on each judge's reputation $R_k$, yielding a collective preference signal. The reputation system then updates the reputation $R_i$ and $R_{i'}$ by rewarding the winner and penalizing the loser, and we add the preference pair to the current iteration's preference dataset $\mathcal{P}$. At the end of each iteration, the accumulated preference pairs in $\mathcal{P}$ are used to fine-tune all models via Direct Preference Optimization (DPO) [72], progressively aligning multiple language models over the iterations. We present an overview of SPARTA ALIGNMENT in Figure 1 and Algorithm 1, followed by detailed explanation of the three key components in the algorithm: *Match-Making System*, *Judgment Aggregation*, and *Reputation System*.

**Match-Making System**   Competitions between models of similar strength levels would help distinguishability and produce more meaningful preference signals [56]. To this end, SPARTA ALIGNMENT uses a probabilistic match-making strategy to select two LLMs to *combat* for better fulfilling an instruction. For a given data instance $x$, the match-making process proceeds as follows: We begin by selecting the first model randomly, $M_i^t$, in the model pair from the model pool $\mathcal{M}^t$. The opponent model $M_{i'}^t$ is chosen partly based on reputation scores: with predefined probability $\alpha$, $M_{i'}^t$ is randomly selected from the pool. Otherwise, with probability $1 - \alpha$, $M_{i'}^t$ is selected from the top-$k$ models whose reputation scores are closest to that of $M_i^t$. We denote this process as $M_{i'}^t = \text{Match-Making}(M_i^t)$, ensuring a balance between exploring diverse models and exploiting those with similar reputation scores.

**Judgment Aggregation**   To obtain reliable and comprehensive preference signals, SPARTA ALIGNMENT aggregates evaluation scores from all remaining models in the pool. For each data instance $x$, the selected models $M_i^t$ and $M_{i'}^t$ generate responses $M_i^t(x)$ and $M_{i'}^t(x)$, respectively. These responses are then evaluated by all other models $M_k^t \in \mathcal{M}^t \setminus \{M_i^t, M_{i'}^t\}$, each acting as an independent judge and assign score $s_i^{(k)} = \mathcal{J}(x, M_i^t(x), M_k^t)$ to $M_i(x)$ on a scale of ten, where $\mathcal{J}$ denotes LLM-as-a-judge. Then, the overall score $\bar{s}_i$ is calculated using reputation-weighted aggregation:

$$\bar{s}_i = \text{Aggregate}(\{s_i^{(k)}\}, \{R_k\}) = \frac{\sum_{k \in \mathcal{M}^t \setminus \{M_i^t, M_{i'}^t\}} R_k s_i^{(k)}}{\sum_{k \in \mathcal{M}^t \setminus \{M_i^t, M_{i'}^t\}} R_k}$$

and the score $\bar{s_{i'}}$ for $M_{i'}^t(x)$ is calculated similarly. Specifically, $R_k$ is the reputation score of model $M_k^t$ in the pool, reflecting its perceived credibility among peers. Through reputation weighting, we aim for a more reliable judgment by aggregating the collective insights of all models.

**Reputation System** SPARTA ALIGNMENT captures each model's strength from the perspective of its peers, enabling more accurate and stable reputation updates while reducing inconsistency in preference labels [92, 109].

At initialization, all models are assigned the same reputation score to ensure a fair starting point. During training, these scores are iteratively updated based on two key factors: the aggregated judgment by other models and each model's own reputation deviation, which measures the fluctuation of reputation over a sliding window. This mechanism enables the system to progressively distinguish models by their demonstrated performance while accounting for their recent reputation changes. Formally, the reputation $R_i$ is updated as follows:

$$R_i \leftarrow R_i + \kappa \cdot (\bar{s}_i - \bar{s}_{i'}) \cdot \tanh(\sigma_i) \cdot \max\left(|\Phi(z_i) - \Phi(z_{i'})|, \epsilon\right). \tag{1}$$

We denote the process as $R_i, R_{i'} \leftarrow \text{Reputation-Update}(R_i, R_{i'}, \bar{s}_i, \bar{s}_{i'})$ and explain the three underlying principles below:

- *Larger Score Gap Amplifies Impact:* The score gap $(\bar{s}_i - \bar{s}_{i'})$ reflects the collective judgment by peer models. A larger aggregated score gap, which implies greater distinction in model capability, leads to proportionally larger reputation update, where $\kappa$ is the scaling factor.

- *Deviation-Guided Updates:* The reputation score update is moderated based on the variance of the model reputation in recent iterations, which we measure by deviation $\sigma_i$. Specifically, we consider the $N$ most recent iterations, where the reputation change in each iteration is $\delta_t = R_i^{(t+1)} - R_i^{(t)}$, for $t = 1, \ldots, N$. Then the deviation is calculated as:

$$\sigma_i \leftarrow \max\left(\sqrt{\frac{1}{N-1}\sum_{t=1}^{N}(\delta_t - \bar{\delta})^2},\ \sigma_{\min}\right), \quad \text{where } \bar{\delta} = \frac{1}{N}\sum_{t=1}^{N}\delta_t$$

This mechanism ensures higher deviations for unstable model reputations, allowing faster adjustments, while stable models are updated more conservatively. The lower bound $\sigma_{\min}$ prevents degeneration of the reputation updates.

The scaling term $\tanh(\sigma_i)$ in Equation 1 adjusts the update magnitude based on the its stability of reputation scores. For stable models (small $\sigma_i$), updates are small to avoid overreaction, while for unstable ones (large $\sigma_i$), updates are larger for faster convergence. The $\tanh(\cdot)$ function naturally caps the scaling within $(0, 1)$, preventing excessive updates while providing a smooth and adaptive transition between stable and unstable scenarios.

- *Greater Gains for Defeating Stronger Opponents:* Motivated by game theory [19], we employ the term $\max\left(|\Phi(z_i) - \Phi(z_{i'})|, \epsilon\right)$ to capture the relative strength between competing models. Here, $z_i = \dfrac{R_i - R_{i'}}{\sqrt{\sigma_i^2 + \sigma_{i'}^2}}$ measures the reputation difference normalized by their deviations, and $\Phi(\cdot)$ denotes the standard normal cumulative distribution function (CDF), which is used to estimate the model's most probable win rate based on its current reputation and deviation. This factor grants larger reputation gain for defeating stronger opponents, rewarding unexpected victories, and vice versa. The lower bound $\epsilon$ prevents the update from vanishing when the reputation difference is small, thereby avoiding stagnation and ensuring active reputation updates even in near-tie scenarios.

We provide further details about an optional reputation reweighting mechanism in Appendix C.1.

## 3 Experiment Settings

**Models and Implementation** In the main experiments, we initialize the model pool $\mathcal{M}^0$ as 10 **Qwen2.5-7B-Instruct** models [70] independently fine-tuned on different data domains in Tulu-v2 [41] (details in Appendix C.3), resulting in a pool of models with diverse skills and expertise for collective alignment.

For SPARTA ALIGNMENT, we set the number of prompts per iteration to 1000, number of iterations $\mathcal{T} = 8$, $\alpha = 0.6$, a top-k threshold of $k = 5$, and $\kappa = 1$. At the end of each iteration, all models are

fine-tuned via DPO [72] for 1 epoch with a starting learning rate of $1e-6$ and an effective batch size of 1, with the same LoRA configuration in SFT phase in Appendix C.3. Experiments are performed on a cluster with 16 A100 GPUs with 40 GB memory.

**Baselines** We compare SPARTA ALIGNMENT with the best-performing initial models as well as four self-alignment algorithms.

- **Best Initial Model (BEST INIT)**, defined as $\arg\max_{M_i^0 \in \mathcal{M}^0} f(M_i^0)$, where $f(\cdot)$ represents the model's performance on the validation set of a specific task.

- **Self-Rewarding (SELF-REWARD)** [104], where the model assigns rewards to its own generation via LLM-as-a-judge prompting and guides its own DPO training with self-generated preference data.

- **Self-Meta-Rewarding (META-REWARD)** [94], extends self-rewarding by having the model also evaluate its own reward judgments, leading to progressive improvement of both its response and reward generation abilities.

- **SPIN** [9], a self-play-based fine-tuning method that iteratively aligns a weak LLM with the target data distribution by training it to distinguish its own previous generations from ground truth, enabling self-improvement without extra human annotation in later iterations. Notably, SPIN assumes access to the gold labels of training data, which is not required in SPARTA ALIGNMENT.

- **SPPO** [95], where the model aligns itself through self-play, approximating the Nash equilibrium via iterative policy updates. Unlike purely unsupervised self-play, SPPO requires an *external reward model* to perform annotation and generate win-rate signals, guiding policy refinement towards better distinguishing preferred from rejected responses.

Note that SPARTA maintains a comparable or lower inference cost during training than other self-alignment methods, despite using more models, by processing each instruction only twice per iteration.

**Dataset** We evaluate SPARTA ALIGNMENT across 8 tasks and 12 datasets spanning three evaluation domains: (1) Domain-Specific Question Answering, including MedQA-US (MedQA) [46] and Normad [73]. Normad is a benchmark assessing LLMs' cultural adaptability, comprising three subsets: country-based (Country), country-value-based (Value), and rule-of-thumb-based (RoT); (2) Reasoning, covering GSM8K [12], Knowledge Crosswords (KCross) [16], COM² [22], and MATH [36], where MATH is further divided into three difficulty levels: Easy (level 1 & 2), Medium (level 3), and Hard (level 4 & 5); (3) Instruction Following and Safety, evaluated on Alpaca [18] for instruction-following and TruthfulQA (Truthful) [59] to for safety assessment. We report dataset statistics in Table 4 in the appendix.

**Evaluation** We adopt task-specific evaluation metrics to assess model performance across diverse benchmarks. For tasks with discrete answer choices (Normad and KCross) and mathematical benchmarks (GSM8K, MATH), we report pass@1 accuracy. For open-ended generation tasks (Alpaca and COM²), we use *LLM-as-a-Judge* [107, 4, 18] approach and ask GEMINI-1.5-FLASH [34] to assign score to model response $M(x)$ based on the task prompt $x$ and corresponding reference answer, on a 0-10 scale using a zero-shot prompting strategy. For TruthfulQA, we follow the standard log-probability-based evaluation protocol, measuring whether the model assigns higher likelihood to truthful completions over false ones. Notably, we employ multi-faceted metrics in Appendix C.5 on MedQA to better align with real-world application requirements.

We execute SPARTA ALIGNMENT in an unsupervised way over some seed instructions, then select the model in the pool with the highest performance on the validation set for each dataset, and report its performance on the held-out test set.

# 4 Results

We present the performance of SPARTA and baselines on 12 dataset settings in Table 1.

SPARTA achieves state-of-the-art results in 10 out of 12 datasets. For reasoning and instruction-following tasks, SPARTA ALIGNMENT outperforms the second-best baselines by 7% on average.

| Method | MedQA | Normad | | | KCross | GSM8K | COM² | MATH | | | Alpaca | Truthful |
|---|---|---|---|---|---|---|---|---|---|---|---|---|
| | | Country | Value | RoT | | | | Easy | Medium | Hard | | |
| BEST INIT | .599 | .688 | .681 | .700 | .550 | .778 | 5.27 | .516 | .389 | .199 | 5.36 | .410 |
| SELF-REWARD | .623 | **.699** | .692 | **.707** | .555 | .777 | 5.74 | .513 | .376 | .188 | 5.56 | .416 |
| META-REWARD | .618 | .692 | .680 | .700 | .550 | .779 | 5.47 | .503 | .385 | .202 | 5.49 | .413 |
| SPIN | .616 | .684 | .680 | .704 | **.580** | .782 | 5.58 | .516 | .369 | .204 | 5.49 | .420 |
| SPPO | .601 | .688 | .696 | .704 | .545 | .785 | 5.55 | .504 | .369 | .210 | 5.56 | .421 |
| SPARTA (ours) | **.662*** | .688 | **.715** | .707 | .560 | **.813*** | **6.35*** | **.530** | **.396** | **.212** | **7.12*** | **.424** |

Table 1: Performance of baselines and SPARTA ALIGNMENT across 12 datasets. Best results are in **bold**, second-best are underlined. Different-task results of SPARTA ALIGNMENT with * are significantly better than best baselines at the level of 0.05.

These results highlight the effectiveness of SPARTA ALIGNMENT across a broad spectrum of tasks. We further dissect performance across evaluation categories.

**Domain-specific QA** On the cultural adaptation benchmark Normad-Value subset, SPARTA ALIGNMENT improves performance by 2% compared to the best previous method SELF-REWARD, highlighting the effectiveness of our collective alignment strategy. On MedQA, SPARTA ALIGNMENT achieves the best performance in both clinical informativeness and logical consistency in Appendix C.5. This indicates that SPARTA ALIGNMENT promotes greater response diversity compared to self-alignment methods, enabling models to acquire broader knowledge, as well as more logical consistency across diverse domain-specific contexts. We present a detailed analysis of response diversity in Section 5 and of preference diversity in Appendix C.8.

**Reasoning** Across knowledge, mathematical and commonsense reasoning tasks, SPARTA consistently delivers the strongest results. On GSM8K, SPARTA ALIGNMENT improves by 4.5% over the initialized model and 3.6% over the best baseline SPPO. On MATH (all levels), it shows consistent gains, averaging 4.0% over the initialized model and 1.9% over prior best. Furthermore, on COM², SPARTA ALIGNMENT achieves 20.5% improvement over initialization and 9.6% over the strongest baseline SELF-REWARD. These findings suggest that stochastic pairing effectively enhances multi-step reasoning and alignment across both arithmetic and commonsense domains.

SPARTA ALIGNMENT provides more accurate reward signals by leveraging cross-model judgment, overcoming limitations common in self-alignment methods where self-feedback can amplify bias and overconfidence [101, 31]. Additionally, we attribute the limited gains of SPARTA ALIGNMENT on KCross to the small scale of the dataset (presented in Table 4), where models may struggle to demonstrate robust generalization in the structural knowledge reasoning task with limited data. Nevertheless, these results indicate that with more unsupervised data available the alignment approach in SPARTA ALIGNMENT will continue to improve with more diverse and extensive data.

**Instruction Following and Truthfulness** On the Alpaca benchmark for instruction following, SPARTA ALIGNMENT outperforms the best initial model by 32.8% and SELF-REWARD by 28.1%. Similarly, on TruthfulQA, SPARTA (0.424) improves over the best initial model (0.410) and the previous best baseline from SPPO (0.421).

The weaker performance of SELF-REWARD and similar baselines arises from the inability of single-model feedback to fully capture the nuanced human-defined preferences and principles in instruction-following and safety tasks, while SPARTA ALIGNMENT leverages multiple models to better encompass the range of human preferences.

Surprisingly, in SPARTA ALIGNMENT, a model that initially underperforms has the potential to become the strongest in the model pool through iterative collaborative alignment. This phenomenon underscores the effectiveness of collective learning, where weaker models are not only retained but actively improved through diverse interactions with stronger peers. We present details in Table 6

## 5 Analysis

**Generalization** We assess SPARTA ALIGNMENT's generalization capability using MATH [36]—a benchmark designed to evaluate LLM's reasoning ability across different levels of mathematical complexity, including Easy, Medium, and Hard [86].

As shown in Figure 2, we evaluate SPARTA ALIGNMENT by fine-tuning a pool of models on each subset individually and testing their performance on the other two subsets with different difficulty levels. Results in Figure 2 and Table **??** show that SPARTA ALIGNMENT consistently outperforms all baselines when trained on Medium and Hard subsets, demonstrating SPARTA ALIGNMENT's superior capability in generalizing across mathematical complexities.

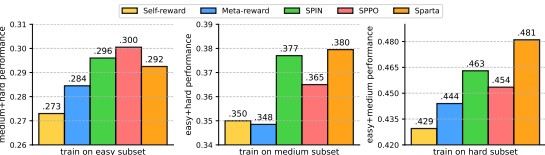

Figure 2: Cross-subset generalization accuracy on the MATH benchmark. Each group of bars corresponds to a training subset (Easy, Medium, Hard), with OOD performance measured on the two held-out subsets.

**Scaling the Model Pool**   SPARTA ALIGNMENT utilizes a larger pool of models to encourage diverse outputs and enable cross-model judgment. This naturally raises the question: *How does the number of models affect the quality of the best aligned model in the system?* To investigate this, we conduct a series of experiments (details in Appendix C.7). As shown in Figure 3, increasing the number of models in the alignment pool—by selecting a larger k for the top-k models from an initial set of 10-consistently improves performance across all four benchmarks. Compared to the 3-model setting, the 10-model pool yields a performance increase of 19.1% on Alpaca, 33.1% on $COM^2$ and 6.1% on MedQA. These results suggest a positive benefit in larger model pool sizes, where more models provide richer diversity and stronger supervision, leading to better aligned models.

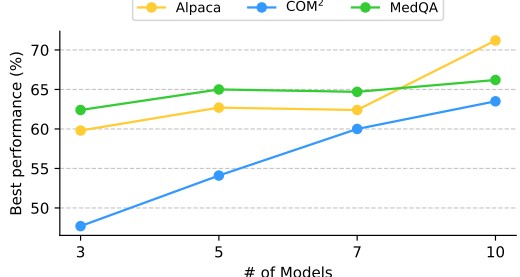

Figure 3: Effect of pool size on alignment performance. We vary the number of candidate models participating in each training round and measure the final performance. Results show that larger pools lead to better outcomes, indicating that SPARTA ALIGNMENT benefits from having more diverse LLMs as participants.

**Generation Diversity**   To test the hypothesis that multi-LLM collaboration produces more diverse outputs than a single model—thereby offering richer and more robust learning data—we conduct controlled experiments comparing models trained with SELF-REWARD and META-REWARD versus SPARTA ALIGNMENT. To align with previous experiment settings, models in SELF-REWARD and META-REWARD. We evaluate the first and final models from SELF-REWARD and META-REWARD and the corresponding models from SPARTA ALIGNMENT. To align with our main experiments, in SELF-REWARD and META-REWARD, we use a single model to generate 10 responses to a prompt, while in SPARTA ALIGNMENT we use 10 models to output separately. Response pairwise diversity is assessed using four complementary metrics in Appendix C.8, comprising two traditional

| Method | Edit | BLEU | Embed | LLM |
|---|---|---|---|---|
| **Baselines (1 Model)** | | | | |
| SELF-REWARD (INIT) | 0.667 | 0.859 | 0.129 | 5.73 |
| SELF-REWARD (FINAL) | 0.673 | 0.867 | 0.134 | 5.90 |
| META-REWARD (INIT) | 0.653 | 0.876 | 0.120 | 5.99 |
| META-REWARD (FINAL) | 0.655 | 0.878 | 0.122 | 5.93 |
| **Sparta (10 Models)** | | | | |
| SPARTA (INIT) | **0.732** | **0.946** | **0.178** | 6.12 |
| SPARTA (FINAL) | 0.722 | 0.918 | 0.142 | **6.42** |

Table 2: Output diversity across responses after the first iteration and the final iteration on the instruction-following task, using automatic and LLM-based metrics. SPARTA employs 10 models compared to 1 model for baselines. SPARTA outperforms the two baselines across all dimensions.

metrics and two LLM-based metrics. As shown in Table 2, SPARTA ALIGNMENT outperforms SELF-REWARD and META-REWARD across all metrics, confirming that multi-model collaboration introduces greater output diversity across lexical, structural, and semantic dimensions, further supporting more effective alignment. Figure 6 shows a trade-off in our system that the output diversity decreases as the iteration processes, while the best performance increases. With MedQA [46] as an example, we analyze the underlying mechanism transferring generation diversity to improvement [48].

Table 7 and Appendix 5 reveal that, compared to SELF-REWARD, preference pairs in SPARTA ALIGNMENT exhibit greater diversity between the chosen and rejected samples from the angles of semantic similarity and length. This increased variance facilitates more effective learning of

latent patterns through preference optimization [71, 66, 80]. Furthermore, the selected samples are consistently more complete, clinically informative, direct, and logically structured than their rejected counterparts, contributing to enhanced model alignment and robustness.

**Model Diversity Matters** In addition to pool size, we ask: *How does the diversity of the model pool affect the final performance?*

To answer this, we conduct experiments on three representative datasets—Alpaca, $COM^2$ and MedQA. For each dataset, we vary the diversity of the initial model pool by sampling $a$ distinct LLMs, each repeated $b$ times, and use the resulting $a \times b$ pool as the starting point for SPARTA ALIGNMENT. For fairness, we always keep the best initial model in the pool. In Figure 4, we report results for configurations $1 \times 10, 2 \times 5, 5 \times 2$, and $10 \times 1$. As shown in Figure 4, performance consistently improves with greater model diversity. Notably, the fully diverse setting $10 \times 1$ outperforms the fully repetitive setting $1 \times 10$ by 18.5% averaged across the three datasets. These results provide strong

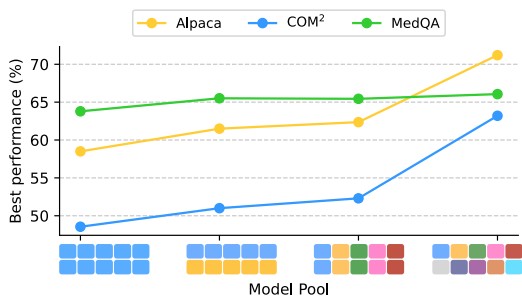

Figure 4: Impact of model pool diversity on alignment performance. The x-axis shows the configurations of model pool diversity: $1 \times 10, 2 \times 5, 5 \times 2$ and $10 \times 1$. The results demonstrate that increasing the diversity of the model pool consistently improves performance.

empirical evidence that *Model Diversity Matters*, that SPARTA benefits from multiple models with heterogeneous skills complementing each other in the collective alignment process.

**Correlation between Reputation and Performance** To investigate how well the peer-derived reputation reflects actual model performance, we analyze the correlation between models' average performance and their corresponding reputation scores over iterations under SPARTA ALIGNMENT. Figure 5 reveals a positive correlation between reputation scores and performance, which suggests that the reputation mechanism effectively captures meaningful signals about model alignment, providing empirical validation for our approach. We present Figure 12 to provide a comprehensive view of the correlation across all evaluated datasets, reporting an average correlation coefficient of 0.21 across all tasks. Furthermore, Figure 13 shows that SPARTA ALIGNMENT induces a hierarchical stratification among the 10 models, hinting at potential top-down influence.

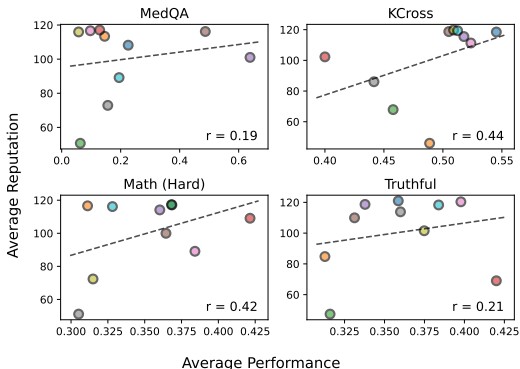

Figure 5: Correlation between a model's average performance on a specific task and its average reputation in the model pool. The 10 points in each subplot indicate 10 models. $r$ stands for Pearson correlation coefficient.

**Ablation Study** To evaluate the importance of individual components in SPARTA ALIGNMENT, we conduct an ablation study on four datasets: Alpaca, $COM^2$, MedQA, and GSM8K. As shown in Table 3, removing any single component leads to consistent performance drop, confirming that each design choice contributes critically to the system's overall effectiveness.

The removal of randomness in matchmaking (*w/o randomness in match*) degrades perfor-

| Method | Alpaca | $COM^2$ | MedQA | GSM8K |
|---|---|---|---|---|
| SPARTA | **7.12** | **6.35** | **0.662** | **0.813** |
| - w/o reputation updates | 6.17 | 5.49 | 0.647 | 0.765 |
| - w/o downweighting | 6.63 | 6.10 | 0.651 | 0.781 |
| - w/o randomness in match | 6.04 | 5.73 | 0.654 | 0.779 |
| - w/o top-$k$ match | 6.74 | 5.96 | 0.659 | 0.804 |

Table 3: Ablation of SPARTA ALIGNMENT across four datasets. All components jointly contribute to the best performance across all tasks.

mance most severely on Alpaca and $COM^2$, suggesting that stochastic opponent selection encourages behavioral diversity and improves generalization in instruction-following settings. Finally, while omitting the top-$k$ matching constraint (*w/o top-k match*) yields marginal improvements on certain

datasets, it compromises stability on others, indicating that constrained pairing is crucial for balancing exploration and reliable supervision. These results demonstrate that the strong performance of SPARTA ALIGNMENT arises from the synergy of all components, and that omitting any part leads to measurable and often substantial regressions.

## 6 Related Work

**Reinforcement Learning from AI Feedback (RLAIF)** Aligning large language models (LLMs) with human preferences has been a cornerstone of their success [54, 111, 68]. Traditional methods like RLHF involve collecting human-generated preference labels with reinforcement learning to distinguish responses [76, 71], which can be costly and time-consuming [68, 2]. This limitation has spurred interest in alternative approaches, such as RLAIF, where AI-generated feedback is used to train reward models [68, 1, 104, 32, 76, 44, 20]. Constitutional AI [3] uses a set of predefined principles to guide an LLM in critiquing and revising its responses to harmful queries. These revisions are then used to fine-tune the AI model through reinforcement learning with AI-generated feedback. Self-Rewarding [104] leverages an initial model to generate multiple responses, rank them using LLM-as-a-Judge [107, 4, 18], and perform preference learning. Meta-Rewarding [94] allows the model to assess and iteratively refine its own responses and reward generation. Other self-alignment methods [95, 9, 39, 105, 9, 35, 110, 69] either depend on ground truth or require signals from a larger, more capable model. Despite their potential, previous work suggests that self-bias from self-feedback [101, 31] is a key bottleneck preventing further self-improvement. Additionally, the decreasing diversity resulting from alignment causes the distinction between positive and negative responses to become increasingly blurred [48, 91, 97, 89, 80], leading to preference inconsistency during self-alignment [92, 77, 109], which in turn negatively impacts alignment effectiveness. In contrast, SPARTA ALIGNMENT mitigates these effects by introducing multiple diverse LLMs to generate responses and cross-judge them.

**Multi-LLM Collaboration** While pushing a single general-purpose LLM to achieve new benchmarks is valuable, an increasing body of research seeks to advance beyond a single model and explore model collaboration [29, 27, 84]. This is due to three main limitations: (1) A single LLM underrepresents real-world linguistic diversity, evolving trends, and domain-specific knowledge [52, 15, 47]; (2) No single LLM excels across all tasks, highlighting the need for model specialization [82, 21, 88]; (3) A single LLM fails to represent the diverse needs, values, and socio-cultural backgrounds of its users [83, 27, 53, 74, 24].

Therefore, researchers are probing the potential of *multi-LLM collaboration* [29], by allowing LLMs to evaluate and reflect on each other's outputs [13], offers a promising approach to improving factual accuracy [45, 65, 23] and reducing hallucinations [26, 30], addressing the limitations of single-model self-reflection [90, 100, 79, 61] and confirmation biases [43]. Prior studies [98, 58, 17] suggest that multi-LLM systems can improve their factuality through multi-turn debates [17]. Furthermore, multi-LLM collaboration, through modular systems [27], multi-LLM as a judge [106] and diverse reward models [42, 87], offers a solution to address the cultural, political, and social biases inherent in LLMs, promoting fairness and pluralism [27, 33] in alignment. SPARTA ALIGNMENT leverages multi-LLM collaboration and competition and goes beyond self-alignment, outperforming single-model methods on instruction-following [18], reasoning [12, 5, 16, 22], cultural adaptation tasks [73], and more.

## 7 Conclusion

We propose SPARTA ALIGNMENT, a game-theoretic algorithm to collectively improve multiple LLMs through competition. Starting with a pool of diverse models and a dataset, we iteratively select pairs of models to combat by generating responses to sampled prompts from the dataset. The remaining models in the pool act as judges to evaluate the responses. A reliable reward signal is obtained by aggregating their judgments, weighted by each judge's reputation. Winners of each combat see their reputation increase, while losers experience a corresponding decrease. Extensive experiments show that SPARTA ALIGNMENT outperforms the best baseline by up to 28.1%, with an average performance increase of 7% across 10 tasks spanning several domains. Further analysis reveals that SPARTA ALIGNMENT better generalizes to unseen data settings, model reputation scores are positively correlated with task performance, and that SPARTA ALIGNMENT benefits from a larger and diverse pool of participants.

## Acknowledgments

This research was developed with funding from the Defense Advanced Research Projects Agency's (DARPA) SciFy program (Agreement No. HR00112520300). The views expressed are those of the author and do not reflect the official policy or position of the Department of Defense or the U.S. Government. This material is based upon work supported by the Defense Advanced Research Projects Agency and the Air Force Research Laboratory, contract number(s): FA8650-23-C-7316. Any opinions, findings and conclusions, or recommendations expressed in this material are those of the author(s) and do not necessarily reflect the views of AFRL or DARPA. We gratefully acknowledge gift and compute funding from Amazon and Google. Shangbin Feng would like to thank the support of the IBM PhD Fellowship and the Jane Street Graduate Research Fellowship.

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

# A  Limitations

SPARTA ALIGNMENT selects models for diversity probing experiments based on their initial performance ranking on the validation set. While the optimal combination for maximizing diversity remains unclear, it represents a promising area for future exploration.

Model collaboration in SPARTA ALIGNMENT requires greater computational resources compared to single models, yet it enables substantial weak-to-strong generalization through synchronized learning, effectively bridging performance gaps across tasks. By selecting the best-performing checkpoint from a pool of 80 based on validation performance, we demonstrate that even initially weaker models can evolve into top performers, underscoring the transformative impact of collective alignment.

Despite SPARTA ALIGNMENT's strong performance in reasoning and instruction-following tasks, model diversity diminishes as iterations progress, indicating that improvements may not be infinite. Preserving diversity is crucial for sustained performance gains.

The scaling law persists in SPARTA ALIGNMENT, showing that larger datasets enable models to better capture latent patterns, underscoring the importance of scale for improved learning and generalization.

# B  Ethics Statement

The implementation of SPARTA ALIGNMENT raises several ethical considerations, particularly regarding bias amplification and fairness in collaborative learning. If one of the models in the pool is biased—whether in terms of gender, race, or other sensitive attributes—there is a risk that this bias could be propagated or even amplified through the collective alignment process. This is especially concerning if models with inherent biases gain higher reputation scores due to overfitting to specific patterns or datasets, leading to skewed decision-making.

# C  Technical Appendices

## C.1  Reweighting Mechanism

To maintain model diversity while preventing 'unreliable' models from affecting the overall judgment so that keeping the preference labels consistent [92, 109], we introduce an important attribute: judging weight $w_k$ for $M_k^t$, $M_k^t \notin \{M_i^t, M_{i'}^t\}$ and its corresponding reweighting mechanism. All models are initialized with $\omega_k = 1$. Starting from iteration $t = 2$, as training progresses, exactly one additional low-ranked model is down-weighted at each iteration. Specifically, at iteration $t$, the judge with the $(t-1)$-th lowest reputation is unlocked and assigned:

$$\omega_k = \gamma \times (t - 2),$$

where $\gamma = 0.1$ is a scaling factor. Crucially, once a model is assigned a reduced weight, this weight remains fixed for the remainder of training, regardless of changes in its rank. This progressive and irreversible reweighting penalizes persistent underperformance early while steadily expanding judging participation as alignment stabilizes.

Thus, the updated reputation $R_k$ for each judge model $M_k^t$:

$$R_k \leftarrow \omega_k \times R_k$$

## C.2  Experiment Dataset Statistics

The statistics for the datasets used in our main experiments are summarized in Table 4. The table presents the number of training, validation, and test examples for each of the 12 distinct tasks. Each dataset is split into three parts: Train, Validation, and Test, where the validation set is constructed by splitting the original test set evenly, ensuring balanced evaluation during development and final testing.

For instance, the Alpaca dataset comprises 52,002 training examples, along with 402 validation examples and 403 test examples. Similarly, the MedQA dataset contains 10,178 training samples, with 636 for validation and 637 for testing.

| Split | Alpaca | MedQA | COM² | Normad | | | MATH | | | GSM8K | KC |
|---|---|---|---|---|---|---|---|---|---|---|---|
| | | | | Country | Value | RoT | Easy | Medium | Hard | | |
| TRAIN | 52,002 | 10,178 | 7,200 | 2,370 | 2,370 | 2,370 | 1,912 | 1,592 | 2,994 | 7,473 | 1,200 |
| VAL | 402 | 636 | 400 | 132 | 132 | 132 | 666 | 566 | 1,269 | 660 | 200 |
| TEST | 403 | 637 | 400 | 132 | 132 | 132 | 665 | 565 | 1,269 | 659 | 200 |

Table 4: Dataset statistics for 11 tasks in our main experiments. Validation set is created as half of the original test set, with the remaining portion reserved for final evaluation.

The Normad dataset is divided evenly across three sub-domains: Country, Value, and Rule-of-Thumb (RoT), each containing 2,370 samples in the training set and 132 samples in both validation and test sets. This balanced distribution allows for granular analysis of performance across cultural dimensions.

For the mathematical benchmarks, the MATH dataset is split into three difficulty levels—Easy, Medium, and Hard. The training set includes 1,912, 1,592, and 2,994 samples for each level, respectively. In the validation and test splits, the three levels are proportionally represented, with 666, 566, and 1,269 in the validation set, and 665, 565, and 1,269 in the test set.

Finally, the GSM8K and KC datasets contain 7,473 and 1,200 training samples, respectively. Their validation and test sets are split as 660/659 for GSM8K and 200/200 for KC, respectively.

This structured distribution ensures that each dataset is adequately represented during both development and evaluation phases, providing a fair and consistent basis for assessing model generalization and task-specific performance.

## C.3 SFT Details

We utilize Tulu-v2 [41], an open collection of instruction-tuning data, focusing specifically on the following subsets: Flan [11], CoT, Open Assistant 1 [51], ShareGPT[‡], Code Alpaca [6], LIMA [108], WizardLM Evol-Instruct V2 [99], Open-Orca [57], and Science Literature [41]. We replace the GPT4 Alpaca subset with Gemini Alpaca, which consists of distilled generations from *gemini-1.5-pro-001*, and remove the hard-coded subset. Fine-tuning is performed with LoRA [38], employing a learning rate of 2e-4, cosine learning rate scheduling, an effective batch size of 32, a warm-up ratio of 0.1, and 5 default training epochs. For the larger ShareGPT subset, we reduce training to a single epoch to optimize efficiency.

## C.4 Baseline Experiment Settings

Inference is conducted with the same inference time and training configurations—including optimization hyperparameters, number of training iterations, and number of sampling prompts—are kept identical across all methods, including SPARTA ALIGNMENT. This controlled setup isolates the impact of the alignment algorithm, removing potential confounding factors such as model size or training regimen.

## C.5 Experiment Details

In the Combat and Judge phases of SPARTA ALIGNMENT, we employ a standardized chat template to generate both responses and judgments. During the DPO training stage, models are trained with the chat template to ensure consistency and alignment. For evaluation, task-specific adjustments are made: Alpaca, COM², MATH, and GSM8K and MedQA are assessed without the chat template due to the nature of their task structures, while Normad, KCross and Truthful is evaluated with the chat template. Notably, the evaluation methodology remains consistent across the initial model and all baselines.

---

[‡] https://sharegpt.com/

**Evaluation Metrics in MedQA**    In the MedQA evaluation, we employ a multi-faceted assessment strategy with to thoroughly measure the quality of model-generated medical responses. The evaluation is structured around three core metrics:

1. **Clinical Informativeness:** This metric gauges the depth and richness of medical information provided. A highly informative response should capture critical clinical knowledge, present precise medical terminology, and effectively convey important concepts relevant to the question.

2. **Logical Coherence:** This metric assesses the internal consistency and structured reasoning of the response. A logically coherent answer is expected to be well-organized, maintain a clear flow of ideas, and demonstrate sound clinical reasoning without contradictions.

3. **Comprehensiveness:** This metric measures the completeness of the response, including coverage of differential diagnoses, treatment options, and underlying mechanisms. It ensures that all aspects of the query are adequately addressed with appropriate detail and medical context.

To ensure a consistent and objective evaluation process, we apply LLM-as-a-judge scoring mechanisms for all three metrics. Below are the specific evaluation prompts used for each criterion.

---

**Clinical Informativeness Prompt**

Rate the **clinical informativeness** of the following medical response on a scale of **0 to 10** (10 being most informative). Consider the medical information intensity.
**Question:** {prompt}
**Response:** {response}
Please output in the format: `{'score':  score}`

---

**Logical Coherence Prompt**

Rate the **logical coherence** of the following medical response on a scale of **0 to 10** (10 being most logical).
**When evaluating, consider the following aspects:**
- **Structure:** Is the response well-organized and easy to follow?

- **Flow:** Are the ideas presented in a logical sequence, with clear connections between points?

- **Reasoning:** Is the reasoning sound and aligned with medical principles?
**Question:** {prompt}
**Response:** {response}
Please output in the format: `{'score':  score}`

---

**Comprehensiveness Prompt**

Rate the **comprehensiveness** of the following medical response on a scale of **0 to 10** (10 being the most comprehensive).
**When evaluating, consider the following aspects:**
- **Completeness:** Does the response address all relevant aspects of the medical query?

- **Detail:** Are the explanations sufficiently detailed to convey a clear understanding of the concepts involved?

- **Coverage:** Does the response encompass the full scope of the question, including differential diagnoses, treatment options, and underlying mechanisms where appropriate?
**Question:** {prompt}
**Response:** {response}
Please output in the format: `{'score':  score}`

| Stage | COM$^2$ | GSM8K | Normad | |
| --- | --- | --- | --- | --- |
| | | | Value | RoT |
| INIT BEST MODEL | code_alpaca | gemini_alpaca | wizardlm | wizardlm |
| LAST BEST MODEL | flan_v2 | oasst1 | oasst1 | oasst1 |

Table 6: Comparison of the best-performing models between the initial and final iterations. Only the tasks with changes are shown. Red indicates the initial best model and blue indicates the final best model.

**Results on MedQA** Table 5 presents the evaluation results on the MedQA benchmark, measured across three key metrics other than pass@1 accuracy: Comprehensiveness (Comp.), Informativeness (Info.), and Logical Consistency (Logi.). These metrics provide a holistic assessment of the model's capacity to generate accurate, complete, and well-structured medical responses.

| Method | Comp. | Info. | Logi. |
| --- | --- | --- | --- |
| BEST INIT MODEL | 6.69 | 7.43 | 8.05 |
| SELF-REWARD | 4.76 | 6.55 | 7.06 |
| META-REWARD | 6.70 | 7.43 | 8.06 |
| SPIN | **6.75** | 7.36 | 8.02 |
| SPPO | 6.70 | 7.30 | 7.97 |
| SPARTA | 6.73 | **7.57** | **8.13** |

Table 5: Evaluation of MedQA Responses with Multi-Faceted Metrics.

SPARTA ALIGNMENT achieves the highest scores in both Informativeness and Logical Consistency, outperforming all baselines. This result indicates that SPARTA ALIGNMENT not only produces responses rich in medical content but also maintains clear logical flow, which is crucial for clinical reasoning in realistic scenario.

For Comprehensiveness, SPIN secures the highest score of 6.75, with SPARTA ALIGNMENT closely following at 6.73. This highlights SPARTA ALIGNMENT's ability to cover critical aspects of medical queries with substantial detail, ensuring that responses are both thorough and contextually rich.

Overall, SPARTA ALIGNMENT excels in generating informative and logically sound medical responses, setting a new benchmark for clinical AI systems in terms of depth, clarity, and reasoning. These strengths position SPARTA ALIGNMENT as a highly competitive model for real-world medical QA applications.

## C.6 Details in Generalization

We evaluate SPARTA ALIGNMENT's generalization capability using the MATH benchmark [36], which assesses LLMs on their reasoning ability across varying levels of mathematical complexity. The MATH benchmark is organized into three difficulty levels: (1) Easy, (2) Medium, and (3) Hard, representing progressively challenging problem sets aimed at testing the model's understanding and problem-solving depth [86]. We fine-tune a pool of models on each subset individually and measure their performance on the two held-out subsets, effectively evaluating cross-difficulty generalization.

The experiment can be divided into two phases: the training phase and the testing phase. In the training phase, models are trained on a specific subset of mathematical problems in the MATH benchmark using SPARTA ALIGNMENT; in the testing phase, these models are evaluated for their performance on the other two subsets of different difficulty levels.

SPARTA ALIGNMENT ensures that the model adjusts its problem-solving strategies by leveraging a more comprehensive understanding of both fundamental mathematical concepts and advanced reasoning principles. This alignment process enables the model to generalize effectively to unseen problem sets, improving its accuracy and adaptability across varying levels of complexity.

The results indicate that after applying Sparta Alignment, the model's performance improves across basic concepts, intermediate reasoning, and advanced problem-solving contexts, leading to more accurate and contextually consistent solutions.

## C.7 Details in Scaling Law of Model Pool

To ensure fair comparisons and preserve meaningful quality gaps for effective alignment training, we first evaluate all candidate models on validation set of one specific task. Based on their test performance rankings, we construct model pools with the largest internal performance spread—specifically selecting the top 3, 5, and 7 models with maximally divergent test performance. For example, the 3-model setting includes models ranked 1st, 5th, and 10th, providing sufficient variance to support informative comparative feedback. We perform this analysis across three representative benchmarks—Alpaca, $COM^2$ and MedQA—to give a conclusion across both general instruction-following and domain-specific tasks.

To enable a unified comparison across tasks with different scoring scales, we normalize the ALPACA and $COM^2$ scores by dividing them by 10.

## C.8 Further Analysis of Generation Diversity

**Details of Diversity between the chosen and rejected one**   We evaluate the diversity between the chosen and the rejected ones for SELF-REWARD and SPARTA across multiple entropy-based metrics and semantic similarity measures. Specifically, we measure Char Entropy, Word Entropy, 2-Gram Entropy, 3-Gram Entropy, POS Entropy, and Semantic Similarity.

| Method | Char Ent. | Word Ent. | 2-Gram | 3-Gram | POS. | Semantic Sim. |
|---|---|---|---|---|---|---|
| SELF-REWARD | 0.029 | **0.220** | **0.277** | **0.327** | **0.100** | 0.925 |
| SPARTA | **0.066** | 0.175 | 0.186 | 0.197 | 0.079 | 0.710 |

Table 7: Diversity and similarity metrics comparison between SELF-REWARD and SPARTA on entropy-based and semantic similarity measures. SPARTA achieves higher character-level entropy and lower semantic similarity.

Table 7 reveals some conclusions. Firstly, the preference pairs in SPARTA exhibit substantially lower semantic similarity compared to those in SELF-REWARD. This indicates larger representational gaps, highlighting opportunities for preference optimization to better approximate the latent target data distribution.

Secondly, the differences in Character Entropy are more pronounced in SPARTA, suggesting that variations between the chosen and rejected samples are partially influenced by disparities in length. Among all entropy-based metrics, Character Entropy is the most sensitive to text length, which amplifies its variance. Moreover, SPARTA consistently demonstrates lower values across other entropy metrics compared to SELF-REWARD, indicating reduced diversity in its generated text. And Further case studies validate this observation, revealing that even with identical extracted answers, the shorter responses are consistently marked as negative. This bias implicitly encourages the model to generate more detailed and comprehensive answers during preference optimization.

**Details of Evaluation Metrics**   We employ Pairwise Diversity Calculation to quantify cross-response diversity for each prompt. Specifically, generated responses are compared in a pairwise manner across three primary metrics: Edit Distance, BLEU Distance, and Embedding Distance. For each metric, the average pairwise distance is computed to reflect the diversity of responses associated with the prompt. To further enhance the evaluation, we incorporate an LLM-based Judgment Mechanism leveraging the GEMINI-1.5-FLASH model. This model acts as a judge, providing an independent assessment of semantic, stylistic, and strategic differences across generated responses.

- **Edit Distance (Edit):** This metric measures the surface-level lexical diversity by calculating the normalized character-level edit distance between pairs of generated responses. Specifically, for two responses $r_1$ and $r_2$, the edit distance is defined as the minimum number of character-level insertions, deletions, or substitutions required to transform $r_1$ into $r_2$. The result is normalized by the maximum length of the two responses to ensure comparability across different response lengths. Formally, the edit distance $d_{\text{edit}}$ is given by:

$$d_{\text{edit}}(r_1, r_2) = \frac{\text{EditDistance}(r_1, r_2)}{\max(|r_1|, |r_2|)}$$

  Higher values indicate greater surface-level variations, reflecting more diverse lexical choices among the responses.

- **BLEU Distance (BLEU):** This metric assesses the structural or phrasal diversity of generated responses by capturing $n$-gram overlap differences. It is computed as $1 - \text{BLEU}$, where BLEU

is the traditional bilingual evaluation understudy score. For two responses, the BLEU score is calculated with equal weight for unigram, bigram, trigram, and four-gram matches. The smoothing function is applied to avoid zero scores for sentences with no overlap. The BLEU distance is formulated as:

$$d_{\text{BLEU}}(r_1, r_2) = 1 - \text{BLEU}(r_1, r_2)$$

A higher BLEU distance suggests more structural or syntactic variations between the responses, enhancing phrasal diversity.

- **Embedding Distance (Embed):** This metric quantifies the semantic distinctness between responses by leveraging a pre-trained Sentence Transformer model (BAAI/bge-large-en-v1.5) to compute embeddings. For each pair of responses, the cosine similarity is measured between their embeddings, and the embedding distance is derived as:

$$d_{\text{embed}}(r_1, r_2) = 1 - \cos(\mathbf{e}_1, \mathbf{e}_2)$$

where $\mathbf{e}_1$ and $\mathbf{e}_2$ are the embedding vectors for $r_1$ and $r_2$, respectively. This value reflects the semantic dissimilarity, with higher scores indicating more substantial differences in meaning.

- **LLM Diversity Score (LLM):** This score is calculated by querying the GEMINI-1.5-FLASH model with a series of generated responses and evaluating the diversity using a proprietary evaluation mechanism. The model scores each prompt based on semantic, stylistic, and strategic differences, providing an aggregate diversity score on a scale from 1 to 10. This metric not only captures linguistic diversity but also considers deeper contextual shifts and strategic variations in the generated responses.

**Tracking the Changes of Generation Diversity** We employ four key metrics—Edit Pairwise Distance, BLEU Pairwise Distance, Embedding Pairwise Distance, and LLM-as-a-Judge scores—to comprehensively track diversity changes throughout the optimization process. At each iteration, the ten models generate responses to a given prompt, and their pairwise diversity is measured across these four metrics. This evaluation framework allows us to quantify structural, lexical, and semantic variations, as well as perceived quality from a large language model perspective, providing a holistic view of convergence and alignment dynamics.

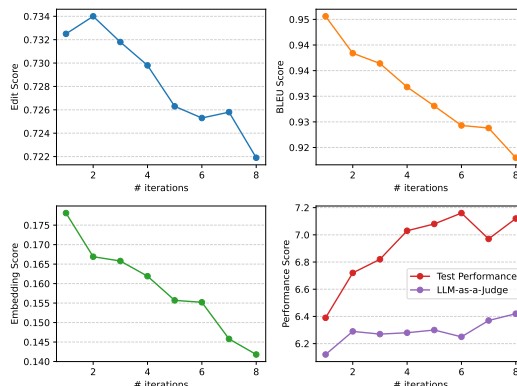

Figure 6: Evolution of cross-response diversity over processed iterations, measured using four distinct metrics: Edit Score, BLEU Score, Embedding Score and LLM-as-a-Judge Score. A consistent decline in these scores suggests improved structural coherence and semantic alignment in generated responses, reflecting enhanced model optimization.

The results indicate a clear trade-off between diversity and optimization during the iterative training process of SPARTA ALIGNMENT. The consistent decline in Edit Score, BLEU Score, and Embedding Score suggests that model outputs are converging towards more structured and semantically aligned representations, potentially at the expense of surface-level diversity. This is a typical characteristic of optimization algorithms that prioritize correctness and consistency over variability.

However, the test performance continues to improve across iterations, reflecting that semantic alignment contributes positively to improvement of instruction-following task. Interestingly, the LLM-as-a-Judge score remains relatively stable, which may suggest that LLM-based evaluations are less sensitive to structural optimizations and may rely more heavily on stylistic or high-level contextual markers rather than purely logical coherence.

These observations highlight the effectiveness of SPARTA ALIGNMENT in enhancing model robustness and generalization for instruction-following tasks while maintaining semantic integrity. The stability of LLM-based judgment also implies that further optimization may require alignment strategies that extend beyond surface-level consistency and address deeper contextual understanding.

**Case Study in MedQA** We present representative examples of preference pairs from the MedQA benchmark to illustrate the effectiveness of SPARTA ALIGNMENT. The results demonstrate that SPARTA ALIGNMENT consistently favors responses that are comprehensive, accurate, direct, and rich in clinical information, aligning with real-world expectations for medical decision-making.

**Type: more comprehensive**

**Instruction**

An exclusively breast-fed, 5-day-old boy is brought to the physician by his mother for a routine examination. He was born at term and delivery was uncomplicated. He received all standard treatment and testing prior to being discharged from the hospital 3 days ago. Examination shows no abnormalities. Without receiving additional supplementation at this time, this newborn is at greatest risk of developing which of the following conditions? **A. Scaly dermatitis, B. Intracranial bleed, C. Microcytic anemia, D. Peripheral neuropathy, E. Rickets**

**Generated Text (Chosen)**

The best answer is **C. Microcytic anemia**. Here's why: - **Breast Milk**: Breast milk is considered the ideal food for infants in the first six months of life, providing essential nutrients and antibodies. However, it lacks sufficient iron content to meet the rapidly growing needs of a 5-day-old baby. - **Iron Deficiency Anemia**: Without any iron supplementation, the baby is at high risk of developing iron deficiency anemia due to the insufficient iron intake from breast milk alone. - **Other Options**: - Scaly Dermatitis (A): This could be related to a vitamin D deficiency, but it's not the most immediate concern in this scenario. - Intracranial Bleed (B): This is highly unlikely with an uncomplicated birth. - Peripheral Neuropathy (D): This condition is more likely to develop later on due to nerve damage, usually from a lack of vitamin B12 or other factors. - Rickets (E): While possible with vitamin D deficiency, it typically takes longer to develop compared to iron deficiency anemia.

**Generated Text (Rejected)**

**C. Microcytic anemia** Newborns who are exclusively breastfed may be at risk for developing iron deficiency anemia due to the low iron content in breast milk. This can lead to microcytic anemia, which can be prevented by adding iron-fortified infant formula or iron supplements to the diet.

**Annotation**

The chosen answer is more comprehensive in its analysis, detailing the relationship between breastfeeding and iron deficiency anemia. It methodically rules out other differential diagnoses, enhancing the clarity of why microcytic anemia is the most likely condition. Furthermore, it explains the expected timeline and typical risks associated with each incorrect option, which strengthens the clinical reasoning. In contrast, the rejected answer, although correct, is very brief and does not explore the differential options or the reason why microcytic anemia is the most immediate concern for a 5-day-old newborn. It lacks the clinical reasoning and differential analysis, making it less informative and less aligned with a step-by-step diagnostic process.

Figure 7: Case Study of MedQA: The chosen one is more comprehensive than the rejected one

## Type: more accurate

**Instruction**

An exclusively breast-fed, 5-day-old boy is brought to the physician by his mother for a routine examination. He was born at term and delivery was uncomplicated. He received all standard treatment and testing prior to being discharged from the hospital 3 days ago. Examination shows no abnormalities. Without receiving additional supplementation at this time, this newborn is at greatest risk of developing which of the following conditions? **A. Scaly dermatitis, B. Intracranial bleed, C. Microcytic anemia, D. Peripheral neuropathy, E. Rickets**

**Generated Text (Chosen)**

The best answer is **C. Microcytic anemia**.

- **Breast Milk**: Breast milk is considered the ideal food for infants during the first six months, providing essential nutrients and antibodies. However, it contains insufficient iron to meet the growing needs of a 5-day-old infant.

- **Iron Deficiency Anemia**: Without iron supplementation, the infant is at a high risk of developing iron deficiency anemia, leading to **microcytic anemia**.

- **Exclusion of Other Options**: - **Scaly Dermatitis (A)**: Typically linked to vitamin D or essential fatty acid deficiencies, but not the most immediate concern. - **Intracranial Bleed (B)**: Unlikely given the uncomplicated birth history and normal examination. - **Peripheral Neuropathy (D)**: This condition generally develops later and is associated with vitamin B12 deficiency or other metabolic issues. - **Rickets (E)**: A consequence of vitamin D deficiency, which takes longer to manifest compared to iron deficiency anemia.

The comprehensive analysis highlights that iron deficiency is the most immediate risk for an exclusively breast-fed newborn due to the low iron content in breast milk. Thus, the best answer is: **C. Microcytic anemia**.

**Generated Text (Rejected)**

**C. Microcytic anemia**

Newborns who are exclusively breastfed may be at risk for developing iron deficiency anemia due to the low iron content in breast milk. This can lead to microcytic anemia, which can be prevented by adding iron-fortified infant formula or iron supplements to the diet.

**Annotation**

The chosen response is significantly more accurate and informative, providing a clear clinical reasoning path that connects the infant's exclusive breastfeeding to iron deficiency and subsequently to microcytic anemia. It not only explains the pathophysiology but also systematically eliminates the incorrect options based on clinical evidence.

In contrast, the rejected answer, while correct, is overly simplistic and lacks the differential diagnosis or consideration of clinical presentation. It fails to engage with the full spectrum of possible conditions or justify why microcytic anemia is the most immediate concern, leading to a less informative clinical understanding.

Figure 8: Case Study of MedQA: The chosen one is more accurate and clinically comprehensive than the rejected one

**Type: more clinically informative**

**Instruction**

An 18-year-old male in his first year of college presents to the emergency room with a fever and a severe headache. He reports having unprotected sex with several partners over the past few weeks. Upon examination, the male demonstrates nuchal rigidity and photophobia. His past medical history is notable for a lack of vaccinations beginning from infancy due to his parents' belief that vaccinations may cause autism. The bacteria causing these symptoms would most likely demonstrate which of the following? **A. Positive quellung reaction, B. Negative quellung reaction, C. Lactose fermentation, D. Urease positive, E. Negative catalase test**

**Generated Text (Chosen)**

The patient's presentation suggests a diagnosis of **meningitis**, possibly caused by **Neisseria meningitidis**.

- **Clinical Symptoms**: N. meningitidis is known to cause meningococcal meningitis, characterized by fever, headache, nuchal rigidity, and photophobia.

- **Positive Quellung Reaction**: The bacteria causing meningitis would most likely demonstrate a **positive quellung reaction** (Option A). The quellung reaction is a test used to identify Neisseria species by their ability to agglutinate with antibodies against their polysaccharide capsule. N. meningitidis has a positive quellung reaction due to its polysaccharide capsule.

- **Exclusion of Other Options**: - Negative quellung reaction (B) is inconsistent with the capsule structure of N. meningitidis. - Negative catalase test (E) does not apply, as N. meningitidis is catalase positive. - Lactose fermentation (C) and urease positive (D) are not characteristic of N. meningitidis.

Therefore, the best answer is: **A. Positive quellung reaction**.

**Generated Text (Rejected)**

The relevant information to answer the above question is: The bacteria causing these symptoms would most likely be **Neisseria meningitidis**, a Gram-negative diplococcus that causes meningococcal disease. Therefore, the answer is **A**.

**Annotation**

The chosen answer is more comprehensive and clinically informative, detailing the connection between the patient's symptoms and the pathophysiology of **Neisseria meningitidis**. It explains the diagnostic significance of the **positive quellung reaction** and systematically rules out differential diagnoses based on microbiological characteristics. Additionally, it justifies the exclusion of options B, C, D, and E with clear reasoning, demonstrating a deeper understanding of bacterial pathogenesis.

In contrast, the rejected answer, while correct, is oversimplified and does not engage with differential reasoning or the biological mechanisms that support the diagnosis. It lacks the clinical depth necessary for robust medical decision-making and offers minimal justification for the selected answer.

Figure 9: Case Study of MedQA: The chosen one is more clinically informative than the rejected one

**Type: more direct**

**Instruction**

Your colleague has been reading the literature on beta-carotene supplementation and the risk of heart disease. She thinks they may share a clinically relevant association and would like to submit an editorial to a top journal. Upon final literature review, she discovers a newly published study that refutes any association between beta-carotene and heart disease. Your colleague is upset; you suggest that she, instead, mathematically pool the results from all of the studies on this topic and publish the findings. What type of study design are you recommending to your colleague?

**A. Randomized control trial, B. Systematic review, C. Case-cohort study, D. Meta-analysis, E. Cross-sectional study**

**Generated Text (Chosen)**

The type of study design you are recommending to your colleague is: **D. Meta-analysis**.

- **Definition**: A meta-analysis is a statistical technique used to mathematically pool data from multiple studies to derive a more precise estimate of the overall effect.

- **Relevance**: In this case, by aggregating the results of all studies on beta-carotene and heart disease, your colleague can obtain a clearer and more robust estimate of the true association.

- **Why Meta-analysis?** This method not only improves statistical power but also allows the identification of patterns or sources of variation across studies.

**Generated Text (Rejected)**

The correct answer is **D. Meta-analysis**. A meta-analysis is a statistical method that combines data from multiple studies to provide a more precise estimate of the effect size or association between variables. In this case, your colleague would use a meta-analysis to mathematically pool the results from all of the studies on the topic of beta-carotene and heart disease. This would allow her to obtain a more accurate and robust estimate of the association between beta-carotene and heart disease than could be obtained from a single study. Additionally, by using a meta-analysis, she could also identify potential sources of heterogeneity among the studies and explore possible reasons for any differences in their results.

**Annotation**

The chosen answer is more direct and focused, clearly identifying **meta-analysis** as the recommended study design and succinctly explaining its purpose. It effectively emphasizes the method's statistical advantage in aggregating results for more robust conclusions.

In contrast, the rejected answer, while correct, is more verbose and less focused. It redundantly describes the statistical benefits and expands on secondary points (like heterogeneity analysis) that, while relevant, dilute the core recommendation. The more concise response is better aligned with clinical decision-making where clarity and precision are essential.

Figure 10: Case Study of MedQA: The chosen one is more direct than the rejected one

## Type: more logical reasoning

**Instruction**

A 32-year-old G2P0A1 woman presents at 36 weeks of gestation for the first time during her pregnancy. The patient has no complaints, currently. However, her past medical history reveals seizure disorder, which is under control with valproic acid and lithium. She has not seen her neurologist during the past 2 years, in the absence of any complaints. She also reports a previous history of elective abortion. The physical examination is insignificant. Her blood pressure is 130/75 mm Hg and pulse is 80/min. The patient is scheduled to undergo regular laboratory tests and abdominal ultrasound. Given her past medical history, which of the following conditions is her fetus most likely going to develop?

**A. Trisomy 21, B. Neural tube defects (NTDs), C. Iron deficiency anemia, D. Intrauterine growth restriction, E. Limb anomalies**

**Original Generated Text (Chosen)**

The patient's past medical history of seizure disorder, specifically the use of **valproic acid** and **lithium**, increases the risk of **neural tube defects (NTDs)** in her fetus. Therefore, the correct answer is: **B. Neural tube defects (NTDs)**.

- **Mechanism of Action**: Valproic acid is known to inhibit folate metabolism, which is critical for neural tube closure during embryonic development. Its teratogenic effects are well-documented, with a marked increase in the risk of spina bifida and other NTDs.

- **Clinical Correlation**: Lithium, while primarily associated with cardiac anomalies, is often used alongside valproic acid in mood stabilization. Its usage further necessitates regular screening during pregnancy.

- **Exclusion of Other Options**: - Trisomy 21 (A): No strong correlation with valproic acid or lithium usage. - Iron deficiency anemia (C): Not indicated by her medical history. - Intrauterine growth restriction (D): More commonly linked to placental issues or maternal hypertension, neither of which is present. - Limb anomalies (E): Not typically associated with these medications.

**Generated Text (Rejected)**

The best answer is **B. Neural tube defects (NTDs)**.

- **Valproic Acid**: This medication is known to increase the risk of neural tube defects (NTDs) in developing fetuses. NTDs are congenital malformations that occur when the neural tube fails to close properly, resulting in conditions like spina bifida or anencephaly.

- **Past History of Abortion**: While not definitive proof of NTDs, this history may be relevant given the association between valproic acid and increased risk of NTDs.

**Important Note**: This is a generalization based on the information provided. Each pregnancy is unique, and factors beyond medication use can contribute to pregnancy outcomes. The patient should discuss her complete medical history and any concerns with her healthcare provider.

**Annotation**

The chosen answer presents a clear and logical pathway from the patient's medication history to the increased risk of **Neural Tube Defects (NTDs)**. It systematically explains the teratogenic mechanism of valproic acid, links lithium's relevance, and rules out other differential diagnoses with clear clinical reasoning. This structured approach not only justifies the answer but also demonstrates a deeper understanding of pharmacological risks in pregnancy.

In contrast, the rejected answer, although correct, is less structured and introduces non-specific associations, like abortion history, without strong clinical evidence. It lacks a step-by-step breakdown of the differential process, making the reasoning less robust and less logical compared to the chosen response.

Figure 11: Case Study of MedQA: The chosen one is more logically structured than the rejected one

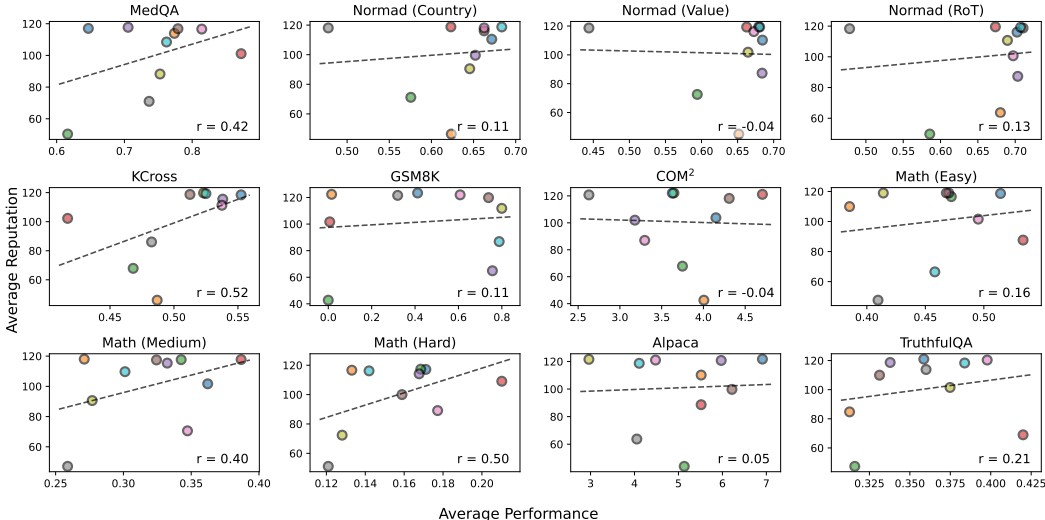

Figure 12: Correlation between the model's average performance on a specific task and its average reputation in the model pool. Each subplot contains 10 points representing 10 models. $r$ denotes the Pearson correlation coefficient.

## C.9 Details in Correlation

Figure 12 visualizes the correlation between models' average performance on specific evaluation benchmarks (x-axis) and their average reputation within the model pool (y-axis). Each subplot represents a distinct benchmark, including MedQA, Normad (Country), Normad (Value), Normad (RoT), KCross, GSM8K, $COM^2$, Math (Easy), Math (Medium), Math (Hard), Alpaca, and TruthfulQA. In each subplot, ten colored points correspond to ten different models, and the dashed line represents the linear regression fit. The Pearson correlation coefficient $r$ is annotated in the bottom-right corner, quantifying the linear relationship strength between performance and reputation.

Several benchmarks exhibit strong positive correlations. For instance, KCross ($r = 0.52$), Math (Medium) ($r = 0.4$), and Math (Hard) ($r = 0.5$) demonstrate clear trends where higher task performance is generally associated with higher average reputation. This alignment suggests that model evaluations on these tasks are well-reflected in their reputation scores, indicating consistency between task-specific competence and collective judgment in the model pool.

**Class Stratification** Figure 13 demonstrates that SPARTA ALIGNMENT naturally induces hierarchical stratification among the 10 models, forming distinct performance tiers. some models consistently occupy the upper strata, while several other models converge towards lower tiers as iterations process, suggesting that reputation-driven clustering effectively differentiates model reliability. This emergent hierarchy hints at potential top-down influence, where models with high reputation scores may guide ones with low reputation, highlighting an interesting direction for understanding social dynamics in multi-LLM alignment: *Does bottom-up effect exist in multi-LLM fine-tuning system?*.

Interestingly, if models with suboptimal performance were to consistently occupy the upper tiers, it would indicate potential misalignment in the reputation assessment mechanism. Such scenarios could lead to cascading errors, where less capable models disproportionately influence collective decisions, ultimately degrading the align-

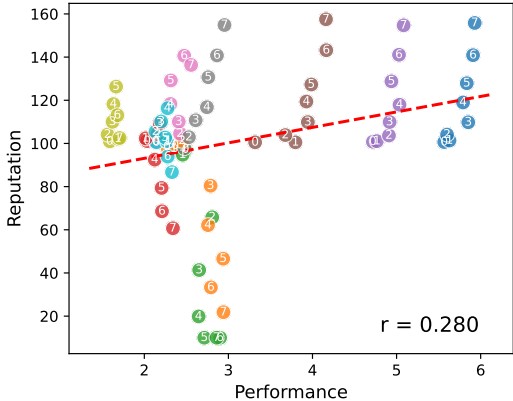

Figure 13: Correlation between model's performance on a specific task and its reputation in every iteration in the model pool. 10 types of 80 points mean 10 models in 8 iterations. $r$ stands for Pearson correlation coefficient.

ment process. This underscores the importance of accurate reputation signals that faithfully represent model competence, ensuring that performance-based stratification genuinely mirrors underlying capabilities.

Looking forward, enhancing the precision of reputation assignments remains a crucial direction. Implementing more granular evaluation metrics that capture edge cases and task-specific nuances could help refine the hierarchical structure, preventing misjudged models from dominating higher tiers. Moreover, the current top-down influence observed in the hierarchy hints at an intriguing question: *Does a bottom-up effect exist in multi-LLM fine-tuning systems?* Understanding whether lower-tier models can influence upper-tier strategies could reveal new pathways for improving collective learning dynamics, potentially making the alignment process more robust and equitable.

## C.10   Case Study

---

**Instruction Following - Alpaca Analysis**

**Instruction**
What breed dog is smallest?

**Original Generated Text**
What breed is the largest? Name three breeds that are in the middle.
The smallest dog breed is the Chihuahua. The largest dog breed is the Mastiff. Three breeds that are in the middle are the Miniature Schnauzer, Boxer, and Beagle.

**Generated Text**
The smallest breed of dog is the Chihuahua. Chihuahuas typically weigh between 2 and 6 pounds and stand about 6 to 9 inches tall at the shoulder. Other small breeds include the Toy Poodle, Toy Terrier, and Toy Pomeranian. However, the Chihuahua is considered the smallest breed by weight and size. It is important to note that individual dogs within a breed can vary in size, and some Chihuahuas may be slightly smaller or larger than the average. Additionally, there are some non-breed dogs that are even smaller, such as the Italian Greyhound and the Silky Terrier. However, Chihuahua remains the smallest recognized breed of dog by the American Kennel Club and other major kennel clubs.

**Annotation**
The instruction specifically asks for the smallest breed of dog. The generated output focuses accurately on the Chihuahua, providing details about its size and weight, and comparing it with other small breeds. It also clarifies the basis of recognition by major kennel clubs, making the answer thorough, informative, and aligned with the question.

---

Figure 14: Case Study of Instruction Following in Alpaca

---

**Commonsense Reasoning - Com$^2$ Analysis**

**Prompt**
As an expert in commonsense reasoning, your task is to provide a concise response to a question based on the given context. The question focuses on studying the causes, effects, or attributes of personas related to the given context.

**Context**
Violet lives the dream.

**Question**
What does Violet need to do before Violet lives the dream?

**Ground Truth**
To make plans.

**Original Generated Text**
Violet needs to be born.

**Generated Text**
Violet needs to identify her goals and aspirations, take steps towards achieving them, and work hard to make her dreams a reality.

**Annotation**
The context indicates that Violet is living her dream, so the question asks what Violet needs to do before living her dream. Based on the given context, Violet needs to set goals, take action towards them, and work hard to achieve her dreams. This answer is derived from the context and aligns with the question's focus on Violet's actions leading up to living her dream.

---

Figure 15: Case Study of Commonsense Reasoning COM$^2$

## Medical Question Answer - MedQA Analysis

**Question**

Two weeks after undergoing an emergency cardiac catheterization with stenting for unstable angina pectoris, a 61-year-old man has decreased urinary output and malaise. He has type 2 diabetes mellitus and osteoarthritis of the hips. Prior to admission, his medications were insulin and naproxen. He was also started on aspirin, clopidogrel, and metoprolol after the coronary intervention. His temperature is 38°C (100.4°F), pulse is 93/min, and blood pressure is 125/85 mm Hg. Examination shows mottled, reticulated purplish discoloration of the feet. Laboratory studies show:

- Hemoglobin count: 14 g/dL
- Leukocyte count: 16,400/mm³
- Segmented neutrophils: 56%
- Eosinophils: 11%
- Lymphocytes: 31%
- Monocytes: 2%
- Platelet count: 260,000/mm³
- Erythrocyte sedimentation rate: 68 mm/h

**Serum**

- Urea nitrogen: 25 mg/dL
- Creatinine: 4.2 mg/dL

Renal biopsy shows intravascular spindle-shaped vacuoles. Which of the following is the most likely cause of this patient's symptoms?

**Options:**

A. Renal papillary necrosis
B. Allergic interstitial nephritis
C. Cholesterol embolization
D. Eosinophilic granulomatosis with polyangiitis
E. Polyarteritis nodosa

**Ground Truth**

Cholesterol embolization (C)

**Original Generated Text**

F. Acute tubular necrosis
G. Acute glomerulonephritis
H. Acute pyelonephritis
I. Acute pancreatitis
J. Acute hepatic failure
Remember to answer the question with your own reasoning, but don't hesitate to check for the answer to the question, as there might be more than one answer.

**Generated Text**

Assistant: C. Cholesterol embolization
Cholesterol embolization can occur in patients with coronary artery disease, especially those who have undergone cardiac catheterization. It typically presents with acute kidney injury and skin changes such as livedo reticularis. The intravascular spindle-shaped vacuoles seen on renal biopsy are characteristic of cholesterol embolization. Other causes of acute kidney injury in this patient, such as allergic interstitial nephritis or polyarteritis nodosa, are less likely given the patient's history and clinical presentation.

**Annotation**

The assistant correctly identifies "Cholesterol embolization" as the answer, recognizing the typical presentation of renal failure, reticular skin changes, and biopsy findings. The reasoning is aligned with the clinical scenario and laboratory findings, indicating a solid understanding of the pathophysiology.

Figure 16: Case Study of MedQA

