# OpenReview forum: "Sparta Alignment: Collectively Aligning Multiple Language Models through Combat"
_NeurIPS.cc/2025/Conference — NeurIPS 2025 poster_

### Official Review · Reviewer_JKyS · 2025-06-25

**Clarity:** 4
**Significance:** 3
**Originality:** 2
**Rating:** 4
**Confidence:** 4

**Summary:**

The paper introduces the idea of aligning multiple large language models through combat. A model and its opponent are selected from a pool of models using a matchmaking strategy. The process consists of three stages: (1) Combat: models generate responses; (2) Judge: other models act as judges; and (3) Update: reputation scores are updated based on the previous stage, and the models are aligned using direct preference optimization. The authors evaluate their method across 12 datasets and compare it against 5 baselines.

**Questions:**

Did the authors explore alternative models for the LLM-as-a-judge component, and if so, did they observe similar trends across their experiments?

**Ethical Concerns:**

["NO or VERY MINOR ethics concerns only"]

**Final Justification:**

**final score**: The paper introduces the idea of aligning multiple large language models through combat. I raised two issues about the bias comparison and extending the results to another model besides Qwen2.5. I believe the paper provided a clear and interesting way of aligning multiple LLMs, and the experimental results section is thorough. The authors addressed my concerns, and based on our discussion, I maintain my overall score of 4. Hence, my final score is **4: Borderline accept**.

**Limitations:**

yes

**Paper Formatting Concerns:**

I didn't find any major issues.

**Quality:**

3

**Strengths And Weaknesses:**

**Strengths:**
- The method is clear and well-motivated.
- The experimental results cover a wide range of datasets, methods, and ablation studies, providing support for the approach.

**Weaknesses:**
- Although the authors mention bias multiple times in their motivation, they do not report any corresponding results, for example, a bias evaluation comparing BEST INIT and SPARTA.
- While the results are extensive and span a range of tasks, they are limited to the Qwen2.5 model. Including at least one additional model—whether open-source or closed-source—would strengthen the findings.

---

> ### Author Rebuttal · Authors · 2025-07-30
>
> We would like to thank the reviewer for their thoughtful comments and feedback.
>
> > Although the authors mention bias multiple times in their motivation, they do not report any corresponding results, for example, a bias evaluation comparing BEST INIT and SPARTA.
>
> We evaluate the judgment bias of BEST INIT and SPARTA-ed model via JudgeBench [1]. From the following table, SPARTA demonstrates a clear advantage in reducing judge bias, achieving the highest accuracy with no invalid responses. This supports our hypothesis that employing collective intelligence of diverse models can mitigate the self-bias [2].
> We leverage JudgeBench [5] to evaluate the judgment bias of models. Specifically, each of the ten models used in Sparta is tasked with independently scoring a set of prompt–answer pairs. Based on the provided preference labels, we report the best accuracy achieved by any single initial model.
> To test our hypothesis regarding collective bias mitigation, we compute the numerical average of the scores from all ten models (excluding invalid scores), and use the resulting aggregate preference to calculate accuracy. This result allows us to assess whether ensembling judgments reduces bias and improves alignment with ground-truth preferences:
> |                    | accuracy % | invalid % |
> |--------------------|------------|-----------|
> | best init          | 40.4       | 0         |
> | average individual | 30.3       | 15.5      |
> | Sparta             | **49.6**   | 0         |
> | improvement        | 9.2        |   /     |
>
> > While the results are extensive and span a range of tasks, they are limited to the Qwen2.5 model. Including at least one additional model—whether open-source or closed-source—would strengthen the findings.
>
> We extend our experiments on another set of models based on gemma-7b-it, and report our results below:
>
> |             | gsm8k     | normad_country | alpaca   |
> |-------------|-----------|----------------|----------|
> | Best Init   | 0.465     | 0.566          | 7.06     |
> | Self-Reward | 0.443     | 0.559          | 6.21     |
> | Meta-Reward | 0.439     | 0.604          | 6.26     |
> | SPIN        | 0.475     | **0.631**      | 7.06     |
> | SPPO        | 0.472     | **0.631**      | 7.07     |
> | Sparta      | **0.486** | 0.612          | **7.23** |
>
> These results confirm that Sparta maintains strong performance even when applied to a different base model. In particular, Sparta achieves the highest scores on GSM8K and alpaca, and remains competitive on normad_country. This supports the generality of our method across both mathematical reasoning and instruction-following tasks, beyond a single model family.
>
> > Did the authors explore alternative models for the LLM-as-a-judge component, and if so, did they observe similar trends across their experiments?
>
> In this work, our primary goal is to explore self-improvement within a multi-model system, where models learn from one another through competitive alignment. As such, we intentionally avoid relying on an external LLM-as-a-judge and instead design a fully internal mechanism, letting models judge each other based on preference signals that emerge from their interactions.
> And for evaluation on open-domained tasks, we try QwQ-32B as an alternative reward model, and find Sparta still outperforms other methods on the alpaca task.
>
> [1] Tan, Sijun, et al. “JUDGEBENCH: A BENCHMARK FOR EVALUATING LLM-BASED JUDGES”, ICLR 2025.
>
> [2]  Xu, Wenda, et al. "Pride and Prejudice: LLM Amplifies Self-Bias in Self-Refinement." ACL 2024.

---

> > ### Comment · Reviewer_JKyS · 2025-08-01
> >
> > Thank you for your rebuttal. Given your thorough response and new experiments, I maintain my original assessment of the paper. The proposed work remains technically solid.

---

### Official Review · Reviewer_CePD · 2025-07-02

**Clarity:** 4
**Significance:** 4
**Originality:** 4
**Rating:** 5
**Confidence:** 4

**Summary:**

Compared to previous approaches that use a single model (i.e., the large model itself) for alignment (i.e., self-alignment), this paper proposes aligning multiple models simultaneously by leveraging the different capabilities of each.
I find this idea quite interesting, and I believe this method could be both simple and effective.

**Questions:**

Is this research the first to propose the idea of aligning multiple large models simultaneously and leveraging their differentiated capabilities?

I believe this is an important question. If so, the authors could emphasize this point in the paper. If not, they should introduce the existing methods.

**Ethical Concerns:**

["NO or VERY MINOR ethics concerns only"]

**Final Justification:**

I have read the authors’ response. My overall opinion remains unchanged. I hope the authors will incorporate all suggestions (including mine and those from other reviewers) that could enhance the overall quality into the final version.

**Limitations:**

Yes.

**Paper Formatting Concerns:**

None.

**Quality:**

4

**Strengths And Weaknesses:**

**Strengths**
1) As mentioned above, overall, I find this idea quite interesting, and I believe this method could be both simple and effective.
2) Further, the writing is clear and straightforward. The writing avoids excessive embellishments, which I appreciate.
3) The experimental results are satisfactory. The code is available as supplementary material.
Additionally, if the authors could make the code publicly available after the paper is accepted, I believe that would be even better.


**Weaknesses**
1) I recommend adjusting the size of some of the figures/tables to make them more visually appealing. For example, reducing the size of Tables 6 and 7.

---

> ### Author Rebuttal · Authors · 2025-07-30
>
> We would like to thank the reviewer for their thoughtful comments and feedback.
>
> > I recommend adjusting the size of some of the figures/tables to make them more visually appealing. For example, reducing the size of Tables 6 and 7.
>
> We agree that visual clarity is important and have carefully considered the sizing of Tables 6 and 7 to ensure readability and consistency with the surrounding text. That said, we are happy to adjust their sizes in the camera-ready version to enhance visual appeal.
>
> > Is this research the first to propose the idea of aligning multiple large models simultaneously and leveraging their differentiated capabilities? I believe this is an important question. If so, the authors could emphasize this point in the paper. If not, they should introduce the existing methods.
>
> To the best of our knowledge, our work is the first to align multiple large models simultaneously and explicitly leverage their differentiated capabilities to generate training pairs for preference learning, enabling self-improvement without relying on gold labels or human-written demonstrations.
>
> Prior works such as [1] and [2] have explored multi-model settings, but their focus remains on improving supervised fine-tuning (SFT) or enhancing human-annotated examples. In contrast, our method harnesses the disagreement and diversity among multiple models to construct preference pairs without external supervision which are then used to improve the models collectively via alignment and competition.
>
> [1] Subramaniam, Vighnesh, et al. "Multiagent Finetuning: Self Improvement with Diverse Reasoning Chains."ICLR 2025.
>
> [2] Wanjia, Zhao, et al. “SiriuS: Self-improving Multi-agent Systems via Bootstrapped Reasoning”, arxiv.

---

> > ### Comment · Reviewer_CePD · 2025-08-07
> > **Comment**
> >
> > Thanks for the rebuttal. I have reviewed it and look forward to seeing more detailed explanations and emphasis in the final version.

---

### Official Review · Reviewer_W7X2 · 2025-07-05

**Clarity:** 3
**Significance:** 2
**Originality:** 2
**Rating:** 4
**Confidence:** 3

**Summary:**

This paper introduces a multi-LLM alignment approach. It can also fall into category of self-improvement since it improves the same model architecture. It evaluates on one Qwen-model and generate multiple instances of it to test their approach. The approach is inspired from "Sparta tribe" and has a three step process. It shows improvement on multiple datasets.

**Questions:**

1. What is the main motivation of this work: self improvement or multi-LLM alignment. I feel the baselines corresponds to self-improvement and are for single LLM. This work is also trying to improve the same model architecture. However, it is being doing by creating multiple instances of the same model. I have two concerns/questions here:
(i) Firstly this work is using 10 instances of the same models in the experiments. However methods such as SPIN, SPPO do not need training of 10 different models. Hence a direct comparison with this work should also take into account the extra computational load. This is not discussed in the paper.
(ii) I also wonder if more simpler approaches like train the model first on the complete training data and then use best-of-N, SPIN or SPPO to make pairs to improve it further.
2. Instead of testing on same model, this work should be tested on different and diverse models.
3. Just testing on Qwen is also not enough. To show generalizability of this method, it should be tested on other models.

**Ethical Concerns:**

["NO or VERY MINOR ethics concerns only"]

**Final Justification:**

Rebuttal has solved most of my concerns. However I still have concerns regarding 7% relative improvement. Improvement seems very minimal.

**Quality:**

3

**Strengths And Weaknesses:**

Weakness:
1. The method of generating preference samples to improve model is intuitive, but the method lack theoretical backing to explain multi-LLM alignment and self-improvement especially a theoretical justification of saturation or optima performance is necessary.
2. It tests on a single model trained to generate different models, however this cannot be compared to multi-LLM system. In reality multi-LLMs are very diverse. Even though they may have similar performance, different LLM architecture have different format of answer generation and may favor different format of answers. It's much simpler to do multi-LLM alignment with same model architecture.
3. SPIN and other baselines mentioned in the paper are not multi-LLM baseline. Multi-LLM baseline should be uses. Even baselines like LLM debate, self-consistency which are only inference time approaches show 5-6% improvement as compared to base models. So such baselines might also be good baselines.

---

> ### Author Rebuttal · Authors · 2025-07-30
>
> We would like to thank the reviewer for their thoughtful comments and feedback.
>
> > The method of generating preference samples to improve model is intuitive, but the method lack theoretical backing to explain multi-LLM alignment and self-improvement especially a theoretical justification of saturation or optima performance is necessary.
>
> While there is theoretical research on LLM alignment [1,2,3,4], their assumptions might not apply to the nuances/complexities in our case.
>
> 1) We are jointly aligning multiple models, while existing theory work focuses on the behavior of a single model. This breaks independence and introduces aspects such as mutual information among models.
>
> 2) Our reward signals (using other LMs as evaluators) are evolving, since all models are iteratively changing in weights and subsequently reward modeling behavior.
>
> While theoretical implications of multi-LLM and self-evolving alignment remains an open research question, we instead pursue empirical evidence to justify the steps in generating preference samples.
> We conduct a comprehensive ablation study across four diverse datasets (Alpaca, COM2, MedQA, and GSM8K) in Table 3, to briefly recap the results:
>
> |                           | Alpaca   | COM$^2$    | MedQA     | GSM8K     |
> |---------------------------|----------|----------|-----------|-----------|
> | Sparta                    | **7.12** | **6.35** | **0.662** | **0.813** |
> | - w/o reputation updates  | 6.17     | 5.49     | 0.647     | 0.765     |
> | - w/o downweighting       | 6.63     | 6.10     | 0.651     | 0.781     |
> | - w/o randomness in match | 6.04     | 5.73     | 0.654     | 0.779     |
> | - w/o tok-k match         | 6.74     | 5.96     | 0.659     | 0.804     |
>
> The results consistently show that removing any individual component of SPARTA ALIGNMENT leads to measurable performance degradation, confirming that the system’s effectiveness arises from the synergy of its modules.
>
> > It tests on a single model trained to generate different models, however this cannot be compared to multi-LLM system. In reality multi-LLMs are very diverse. Even though they may have similar performance, different LLM architecture have different format of answer generation and may favor different format of answers.
>
> We additionally test out Sparta Alignment with a pool of heterogeneous model architectures, specifically 3 models fine-tuned from gemma-7b-it (code alpaca, cot and flan_v2) and 3 models fine-tuned from qwen2.5-7b-instruct (gemini_alpaca, lima, oasst1). We report the performance of best initial gemma-based models and Sparta-ed gemma-based models on three tasks:
>
> |           | gsm8k     | alpaca   | normad_value |
> |-----------|-----------|----------|--------------|
> | best init | 0.465     | 4.84     | 0.566        |
> | 2*3       | **0.475** | **5.29** | **0.578**    |
>
> Results show that Sparta Alignment is compatible with heterogeneous model pools. These results demonstrate that Sparta Alignment is compatible with heterogeneous model architectures, maintaining strong performance across both reasoning and instruction-following tasks. This supports the generality of our framework and its applicability in settings where model diversity is the norm rather than the exception. As this is a quick test with limited compute, we expect more models, architectures, and self-alignment iterations would further boost performance.
>
> > SPIN and other baselines mentioned in the paper are not multi-LLM baseline. Multi-LLM baseline should be uses
>
> While inference-time techniques such as LLM debate and self-consistency may show gains during multi-LLM inference, our work targets a training-time approach with self-improvement setting, where multiple models interact to generate preference data and update model parameters through training. In contrast, methods like LLM debate and self-consistency operate purely at inference time and do not involve any model updates.
>
> Therefore, while methods like SPIN are more comparable in terms of their training-based objective, LLM debate and self-consistency differ fundamentally in that they do not perform any model updates, and thus address a different goal. That said, we view these methods as complementary—SPARTA ALIGNMENT can potentially benefit from inference-time enhancements as well, and integrating the two is an exciting direction for future work.
>
> Importantly, to the best of our knowledge, this is the first work that applies preference learning in a multi-model self-improvement framework. As such, existing multi-LLM baselines—whether inference-only or alignment-focused—do not fully align with our setup. This makes direct baseline selection nontrivial, and we have chosen the most relevant training-based methods currently available.
>
> Nevertheless, we additionally compare with two multi-LLM non-training baselines, self-consistency and multi-model debate.
>
> For self-consistency, we report best majority voting accuracy over 10 samples (maj@10) from the best init model:
> |              | gsm8k     | math_easy | normad_value |
> |--------------|-----------|-----------|--------------|
> | Self-consist | 0.779     | **0.568** | 0.681        |
> | Sparta       | **0.813** | 0.530     | **0.715**    |
>
> Sparta outperforms Self-consistency on gsm8k and normad_value tasks.
>
> Due to context length limits, we use 9 models to participate in the debate and 1 model to summarize the content. All 10 models are identical to those used in the initial Sparta setup. For fair comparison, we run 8 rounds of debate and report the best performance:
>
> |        | math_easy | alpaca   | normad_value |
> |--------|-----------|----------|--------------|
> | debate | 0.317     | 8.44     | 0.681        |
> | Sparta | **0.530** | **9.38** | **0.715**    |
>
> Across all three tasks, our proposed Sparta method consistently outperforms the debate baseline. This demonstrates the effectiveness of Sparta in collaborative self-improvement, especially under subjective and open-ended evaluation settings.
>
> > What is the main motivation of this work: self improvement or multi-LLM alignment. I feel the baselines corresponds to self-improvement and are for single LLM. This work is also trying to improve the same model architecture.
>
> The main motivation is to improve self-alignment, while multi-LLM is only a means to the end, specifically by leveraging a competitive interaction strategy among multiple models. Rather than improving a single LLM in isolation, our method facilitates collective self-improvement within a multi-agent system, where models co-evolve by learning from and challenging each other. As such, much of the baselines and evaluation focus on self-alignment methods and evaluation.
>
> > Firstly this work is using 10 instances of the same models in the experiments. However methods such as SPIN, SPPO do not need training of 10 different models. Hence a direct comparison with this work should also take into account the extra computational load.
>
> We would like to clarify that we have explicitly discussed the computational implications in both the main text and the limitations section. Specifically, in lines 159–161, we compare inference costs and show that SPARTA requires equal or even lower inference time than other baselines under our experimental setting. Additionally, in lines 774–775, we clearly acknowledge that SPARTA ALIGNMENT involves greater computational resources due to multi-model collaboration, which we frame as a trade-off for enhanced alignment dynamics and substantial weak-to-strong generalization, that is to say, weak models at first can become the most powerful models finally.
>
> > I also wonder if more simpler approaches like train the model first on the complete training data and then use best-of-N, SPIN or SPPO to make pairs to improve it further.
>
> Our work targets at a realistic setting where high quality ground truth or preference labels are hard to get at scale. We would like to clarify that our method does not rely on external supervision and do not assume access to external reward models, ground truths, etc. Instead, all preference data is generated and judged dynamically by the pool of models that are collectively evolving. So we can’t “train the model first” as there is no training data available.
>
> Our approach enables emergent self-improvement through multi-model interaction, where models co-evolve via competitive alignment and collective feedback. This dynamic interaction helps mitigate biases (e.g., from a fixed judge) and improves robustness without relying on predefined external metrics.
>
> We believe this shift from single-model fine-tuning to multi-agent alignment-based self-improvement is a key contribution of our work, and we view it as complementary rather than redundant to existing simpler methods.
>
> > this work should be tested on different and diverse models.
>
> We extend our experiments on another set of diverse SFT models based on gemma-7b-it, and report our results below:
>
> |             | gsm8k     | normad_country | alpaca   |
> |-------------|-----------|----------------|----------|
> | Best Init   | 0.465     | 0.566          | 7.06     |
> | Self-Reward | 0.443     | 0.559          | 6.21     |
> | Meta-Reward | 0.439     | 0.604          | 6.26     |
> | SPIN        | 0.475     | **0.631**      | 7.06     |
> | SPPO        | 0.472     | **0.631**      | 7.07     |
> | Sparta      | **0.486** | 0.612          | **7.23** |
>
> These results confirm that Sparta maintains strong performance even when applied to a different model architecture.
>
> [1] Rafailov, Rafael, et al “Direct Preference Optimization: Your Language Model is Secretly a Reward Model”, NeurIPS 2023.
>
> [2] Wu, Yue, et al “Self-Play Preference Optimization for Language Model Alignment”, ICLR 2025
>
> [3] Meng, Yu, et al “SimPO: Simple Preference Optimization with a Reference-Free Reward”,  NeurIPS 2024.
>
> [4] Shao, Zhihong et al “DeepSeekMath: Pushing the Limits of Mathematical Reasoning in Open Language Models”, arxiv.

---

> > ### Comment · Reviewer_W7X2 · 2025-08-05
> >
> > Thanks for the nice and detailed rebuttal. I have two more clarification questions.
> >
> > 1. The percentage improvement of 7% is relative or absolute ? While checking table 1, I see not significant improvement sometime ~1%.
> > 2. While the original performance of Qwen2.5-7B-Instruct on gsm8k is around 91%. Does training on Tulu-v2 reduces the performance ? (as can be seen in the first row of Table 1)

---

> > > ### Author Response · Authors · 2025-08-05
> > >
> > > Thank you for following up!
> > >
> > > 1. We assume it is the 7% on line 186. It is relative improvement. In addition to improvement percentage, we also perform t-test and report "Sparta Alignment with * are significantly better than best baselines at the level of 0.05." (Table 1, caption).
> > >
> > > 2. Tulu-v2 features 10 data domains [1]: flan, open assistant, sharegpt, gpt4-alpaca, code-alpaca, lima, wizardlm, open-orca, hardcoded, and science, none of them specifically about math, so some drops are expected. While the original qwen checkpoint might be hyper-optimized for math datasets, we employ Tulu v2 to create a general-purpose pool of model skills and specialization.
> > >
> > > We assume the "around 91%" number is quoted from their technical report: in there it was evaluated with 4-shot prompting (page 8 in [2]) while our evaluation is zero-shot, so performance numbers here and there are not directly comparable.
> > >
> > > [1] Ivison, Hamish, et al. "Camels in a changing climate: Enhancing lm adaptation with tulu 2."
> > >
> > > [2] "Qwen2.5 Technical Report."

---

### Official Review · Reviewer_mK6c · 2025-07-05

**Clarity:** 4
**Significance:** 3
**Originality:** 4
**Rating:** 5
**Confidence:** 2

**Summary:**

The authors propose a novel preference optimization method that aligns multiple LLMs through competition and combat, effectively addressing the self-bias and limited generation diversity issues inherent in single-model self-alignment. Extensive experiments across three evaluation domains, eight tasks, and twelve datasets demonstrate that the proposed method consistently outperforms single-model baselines. The paper is well-written and well-organized, and the reported performance improvements are impressive.

**Questions:**

1.	Could you provide detailed compute cost estimates (e.g., GPU hours, memory usage) compared to self-alignment methods?
2.	Given that LLMs act as judges for each other, is there a risk of reinforcing shared biases? Is it possible to foresee and avoid this?
3.	How might this method be used in practice?
4.	Can this method effectively scale to larger models? Since larger models might generate more diverse and less biased outputs, would this affect the system's dynamics?

**Ethical Concerns:**

["NO or VERY MINOR ethics concerns only"]

**Final Justification:**

I have read the author's rebuttal and I believe it is consistent with my expectations, so I have maintained my score.

**Limitations:**

yes

**Quality:**

3

**Strengths And Weaknesses:**

**Strengths**

1.	The combat system and reputation system seem novel and well-motivated. The design choices in these systems are theoretically sound and practically justified.
2.	The authors provide detailed ablation studies on the size of the model pool, generation diversity, the importance of model diversity, and the correlation between reputation and performance. The ablation studies provide strong evidence to support the contributions of each component.
3.	The proposed method significantly outperforms strong self-alignment baselines across a wide range of benchmarks, including challenging reasoning tasks.

**Weakness**

1.	The approach requires maintaining and fine-tuning multiple large models in parallel, which may be infeasible for many labs or practical deployment scenarios.
2.	The proposed method can be further verified in a real-world test.
3.	Using peer models as judges might risk reinforcing collective biases, especially if all initial models share certain weaknesses or cultural biases.

---

> ### Author Rebuttal · Authors · 2025-07-30
>
> We would like to thank the reviewer for their thoughtful comments and feedback.
>
> > The approach requires maintaining and fine-tuning multiple large models in parallel, which may be infeasible for many labs or practical deployment scenarios.
>
> While our default setup uses parallelization to accelerate the inference and alignment process on 5 GPUs, Sparta Alignment can also run on a single 40GB GPU by serializing the computation. Our implementation supports any (# model, # GPU) combinations.
>
> Empirically, in our main experiments, SPARTA runs on 5 40GB GPUs for around 6 hours for training 10 models with 8 iterations. We personally would’t call 5 GPU for 6 hours “infeasible” amount of compute.
>
> > The proposed method can be further verified in a real-world test.
>
> While our current experiments are conducted in controlled settings, we have carefully designed them to closely reflect practical usage scenarios and challenges. We agree that deployment in real-world environments would be a valuable next step and plan to pursue this direction in future work. If you have a specific type of real-world test in mind, we would be grateful for further clarification to better align with those expectations.
>
> > Using peer models as judges might risk reinforcing collective biases, especially if all initial models share certain weaknesses or cultural biases. Given that LLMs act as judges for each other, is there a risk of reinforcing shared biases? Is it possible to foresee and avoid this?
>
> Thank you for the insightful comment. We agree that using peer models as judges may risk reinforcing shared biases, a challenge that is not unique to our method but common across all existing self-improvement frameworks that rely on LLM-as-a-judge.
> To mitigate this issue, we deliberately employ ten diverse LLM instances with differentiated capabilities. This diversity enables a richer spectrum of perspectives in the evaluation process, reducing the likelihood that a single dominant bias will propagate unchecked.This design draws on the Language Model Council framework [1], which shows that a pluralistic set of judges leads to more robust, human-aligned, and bias-resistant evaluations.
>
> This idea also resonates with the philosophy behind Modular Pluralism [2] which advocates for representational diversity in alignment by incorporating community-specific language models. While Modular Pluralism focuses on pluralistic human values, our approach similarly embraces model-level pluralism to foster more balanced judgment dynamics within a self-improving system.
>
> > Could you provide detailed compute cost estimates (e.g., GPU hours, memory usage) compared to self-alignment methods?
>
> For parallel structure, we use 5 40GB GPU to run about 7 hours to finish the whole process with 10 models with transformers to roll out. For SPIN and SPPO, the whole process lasts for 8 hours: 1 40GB GPU for rolling out responses and 4 40 GB GPU for training models; for self-rewarding, we use a single 40 GB GPU to run about 8 hours; and for meta-rewarding, we use a single 40 GB GPU to run about 20 hours.
>
> > How might this method be used in practice?
>
> In real-world deployment settings where human evaluation is costly or infeasible at scale, our method allows a system of models to self-improve through structured multi-agent interaction. By enabling models to critique, compete, and align with one another, we effectively offload the supervisory burden from humans to an automated process—offering a path toward scalable, low-cost, and high-quality oversight. Furthermore, while self-alignment based methods tend to converge early and then plateau, SPARTA provides more consistent improvement over iterations.
>
> > Can this method effectively scale to larger models? Since larger models might generate more diverse and less biased outputs, would this affect the system's dynamics?
>
> Yes, our method can effectively scale to larger models. In principle, the alignment and self-improvement dynamics are model-agnostic and can be applied to stronger or larger base models. Practically, we can develop more efficient multi-GPU parallelization strategies to support larger model sizes without compromising interaction throughput if needed.
>
> [1] Zhao, Justin, et al. “Language Model Council: Democratically Benchmarking Foundation Models on Highly Subjective Tasks”, NAACL 2025.
>
> [2] Feng, Shangbin, et al. "Modular Pluralism: Pluralistic Alignment via Multi-LLM Collaboration" EMNLP 2024.

---

> > ### Comment · Reviewer_mK6c · 2025-08-04
> >
> > Thank you for your response, and I maintain my score.

---

### Comment · Area_Chair_mVPf · 2025-08-05
**Author-Reviewer Discussions**

Dear Reviewer W7X2/CePD,

Thanks for your review! The authors have provided their rebuttal. Please respond to the authors and update your review as appropriate.

Thank you!
AC

---

### Decision · Program_Chairs · 2025-09-17

**Decision:**

Accept (poster)

**Comment:**

The paper proposes Sparta Alignment, a new method to align multiple LLMs through competition and combat. The framework is executed in an iterative manner, where multiple LLMs compete against each other in completing instructions while serving as judges for the other competitions. For each iteration, one instruction and two models are selected for a duel, the other models evaluate the two responses, with the evaluation scores aggregated through a ranking system.

All the reviewers found the idea to be novel and interesting, and the empirical evaluation to be solid and convincing. A few concerns, such as the use of a single model family, were raised in the initial reviews. The authors have provided satisfactory responses to address these concerns.

After the rebuttal, all reviewers have agreed to accept this paper, and believe its contribution will be valuable to the community.